# Structure of AcMNPV nucleocapsid reveals DNA portal organization and packaging apparatus of circular dsDNA baculovirus

Gregory Effantin [1] ✉, Eaazhisai Kandiah [2] ✉ & Martin Pelosse [3] ✉

Baculoviruses are large DNA viruses found in nature propagating amongst insects and lepidoptera in particular. They have been studied for decades and are nowadays considered as invaluable biotechnology tools used as biopesticides, recombinant expression systems or delivery vehicle for gene therapy. However, little is known about the baculovirus nucleocapsid assembly at a molecular level. Here, we solve the whole structure of the *Autographa californica* multiple nucleopolyhedrovirus (AcMNPV) nucleocapsid by applying cryo-electron microscopy (CryoEM) combined with de novo modelling and Alphafold predictions. Our structure completes prior observations and elucidates the intricate architecture of the apical cap, unravelling the organization of a DNA portal featuring intriguing symmetry mismatches between its core and vertex. The core, closing the capsid at the apex, holds two DNA helices of the viral genome tethered to Ac54 proteins. Different symmetry components at the apical cap and basal structure are constituted of the same building block, made of Ac101/Ac144, proving the versatility of this modular pair. The crown forming the portal vertex displays a C21 symmetry and contains, amongst others, the motor-like protein Ac66. Our findings support the viral portal to be involved in DNA packaging, probably in conjunction with other parts of a larger DNA packaging apparatus.

Baculoviruses are a family of large circular double-stranded DNA (dsDNA) virus ranging from 80 to 180kbp[1] infecting predominantly insects. As such, they are naturally involved in the regulation of insect population and are applied extensively in agronomy as bioagent for pest control[2,3], reducing the negative impact of chemical insecticides. Moreover, baculovirus expression vector system (BEVS) is widely deployed as biotechnological tools for complex recombinant protein expression in insect cell culture thanks to their large cargo capacity[4–6]. Such capacity combined with the possibility of displaying foreign proteins on the surface of the virion convey baculoviruses to also become essential tools for gene therapy and vaccine development[7–9].

The most well-studied baculovirus is *Autographa californica* multiple nucleopolyhedrovirus (AcMNPV). Its 134 kbp dsDNA genome was sequenced and shown encoding for 155 proteins. As for other members of the baculoviridae family, the AcMNPV infection cycle comprises two types of infectious virions: the occlusion-derived virion (ODV) and budded virion (BV). Both ODV and BV are reported to contain the same nucleocapsid and their distinct phenotypes differ mostly by their envelope composition. In nature, ODVs prime infection of the host midgut upon occlusion bodies oral ingestion whereas BVs are responsible for systemic infection through entering of many cells. It is only the BV form, despite being more fragile than ODV, that is extensively used in biotechnology.

[1]Univ. Grenoble Alpes, CNRS, CEA, Institut de Biologie Structurale (IBS), 38000 Grenoble, France. [2]European Synchrotron Radiation Facility (ESRF), 71 Avenue des Martyrs, 38000 Grenoble, France. [3]European Molecular Biology Laboratory, 71 Avenue des Martyrs, CS 90181, 38042 Grenoble, Cedex, France. ✉e-mail: gregory.effantin@ibs.fr; eaazhisai.kandiah@esrf.fr; mpelosse@embl.fr

However, very little is known about baculovirus capsid assembly mechanisms and structural organization. Baculoviruses have their large circular dsDNA genome packaged into an enveloped rod-shaped nucleocapsid. Very recently some studies reported for the first-time high-resolution structures of the capsid (sheath, part of the apical cap and basal structure) obtained from overexpressed capsid protein (VP39)[10] or purified ODV from dead larvae[11]. These structures support previous observations of baculovirus morphology containing a sheath and two cap structures[12] and partially identify the structural composition of the nucleocapsid. The sheath is exclusively constituted of a helical assembly of VP39. The apical and the basal structures were shown to be made of Ac104, Ac144, Ac101, Ac109, Ac142 and VP39 organized as asymmetric units within similar C14 rings, with Ac98 being solely at the base. Onto the C14 basal ring sits a C7 plug made of Ac144/Ac101 assemblies and sealing the nucleocapsid. By contrast the apical C14 ring was reported harboring a cap for which the exact nature and symmetry could not be addressed. This cap was suggested as hypothetically containing a portal complex reminiscent of those found in other dsDNA viruses[11].

In contrast to small viruses for which virions assembly often relies on genome (DNA or RNA) coating by a nucleoprotein, baculoviruses are large viruses and energy-dependent mechanisms are very likely required for DNA pumping into preformed nucleocapsid[13], similar to bacteriophages and herpes viruses[14–16]. However, until now no evidence is available regarding the existence of such energy-driven motor and only a speculative role of Ac66 as being involved was proposed[1] due to its homology with motor proteins.

Herein, we report high-resolution cryo-electron microscopy (cryo-EM) structures of the BV nucleocapsid combined with de novo modeling and Alphafold predictions. Our results confirm prior observation on the helical sheath and the basal structure, and provide new insights into the apical cap. We determined its complete structure, identified new components in its C14 ring and unveiled the apical cap symmetry mismatch as well as the dsDNA portal protein composition. We determined and assigned extra densities previously overlooked in the apical C14 ring anchor-1 as being PTP (Protein Tyrosine Phosphatase) and Ac66. Our structures determine the inner portal as being composed by a C2 symmetric plug, unambiguously harboring two dsDNA helixes, onto which sits a crown-shaped structure having a C21 symmetry. We determined the C2 plug as being composed of Ac101, Ac144 and DNA-bound Ac54 whereas the C21 crown is made of Ac66, Ac101 and Ac102.

This newly identified DNA portal of AcMNPV is the first one described to date amongst the whole baculoviridae taxon and also for circular double-stranded DNA viruses. The full architecture of the infectious AcMNPV baculovirus budded virions (BV) propagating among insect cell cultures, reported here, contributes to the structural landscape of the baculovirus, currently a less complete database for a virus that is extensively used in biotechnology.

## Results

### Structural determination of the native AcMNPV nucleocapsid

To determine the cryo-EM structure of the intact DNA translocation portal of AcMNPV, we extracted intact BVs by a single concentration step to the medium of our infected insect cells, aiming to retain their structural integrity and deliberate encapsulation to capture their functional form as faithfully as possible. The differences we observe with recently disclosed structures[10,11] on purified empty nucleocapsids or ODV can be due to differences in sample purification or still unknown maturation steps rendering BV different from ODV. Studies involving mass spectrometry analysis[17] revealed differences in protein composition between BV and ODV, relative to the individual envelope for BV and its content.

Cryo-EM of purified BVs revealed DNA-filled or -empty nucleocapsids (Supplementary Fig. 1A). In this study, we focused on filled BV particles (termed BV from now on) which constitute the vast majority of the nucleocapsids in the sample. Each BV particle shows the nucleocapsid harbors two different and huge cap structures on both poles, which are termed basal structure and apical cap in line with previous publications[1,11,18]. A membrane envelope encompasses elegantly the BV particle, touching both the basal and the apical structures (Fig. 1A and Supplementary Fig. 1A). The glycoprotein, GP64, is clearly visible attached to the envelope at the two poles of the virion where the basal and apical ends locate.

### Single particle cryo-EM reconstruction of the nucleocapsid

The BV nucleocapsid structure consists of three parts, a central helical sheath containing the viral genome, an apical cap and a basal structure attached to both extremities of the helical sheath (Fig. 1). To delineate the structure of these individual components tailored by nature to form different symmetries, a total of 8 cryo-EM volumes have been reconstructed (see Material and Methods and Supplementary Fig. 2, Fig. 3, Fig 7, Fig 8 and Table 1). The central tubular region is helical with an axial rise of 42.6 Å and a twist of −7.1° and C14 symmetric. The 3D reconstruction reached an overall resolution of 3 Å (Fig. 1A, Supplementary Fig. 1B–E, G–I) and is composed of the capsid protein, VP39, which in its dimeric nature (Supplementary Fig. 1F), generates a left-handed helix. The entire basal structure was determined to 5.2 Å resolution (applying a C7 symmetry) (Supplementary Fig. 4A–C), delineating two different symmetrical rings (Fig. 1C): an outer one with C14 symmetry surrounding an inner one with C7 symmetry (Supplementary Fig. 4 D–G). 3D refinements focused on these individual rings with their respective symmetries applied, resulted in higher resolution structures, 4.1 Å for C14 (Supplementary Fig. 5A, B and F) and 3.4 Å for C7 rings (Supplementary Fig. 6A–D). The apical cap at the opposite end of the BV nucleocapsid exhibits a more intricate symmetrical arrangement that can be delineated by three oligomeric rings having different symmetries. As opposed to the basal structure which follows overall a C7 symmetry, the apical one, in its entirety, is asymmetric with no common rotational symmetry. A clarified reconstruction could be obtained for the apical cap by performing focused 3D refinements on the various rings (Fig. 1B). The bulk at the core of the apical cap that effectively seals the nucleocapsid has an overall C2 symmetry (Supplementary Fig. 9) and was solved to 6 Å. It consists of an outermost C14 ring solved by focused refinement, at 4.7 Å resolution (Supplementary Fig. 10A, B and D), encapsulating an innermost plug (part of the portal complex) that has C2 symmetry, solved at a resolution of 6.1 Å (Supplementary Fig. 11A–H). Finally, directly above this core part, lies another crown-like structure, having a C21 symmetry which was solved to 3.5 Å (Supplementary Fig. 12A–C). Together with the C2 plug, the C21 ring forms the portal complex of BV. As a highlighting feature, the innermost C2 part of the portal reveals undisputable densities for two dsDNA strands (Fig. 1A and Supplementary Fig. 11B, I).

In order to comprehend the whole architecture of the basal, the capsid and the apical structures, we built atomic models for each asymmetric unit of these individual symmetric components. A single stand-alone program including Alphafold[19] was not sufficient to catalogue all the individual densities. To identify the proteins arranging the apical and basal complexes, we performed atomic model building experiments using an integrated workflow employing various programs as described in the method section. Our mixed approach allowed the identification of all the proteins composing the symmetric parts of both the basal and apical caps of BVs. This work contributed finally to the confirmation of the presence of 8 proteins identified in a previous study[11] and to the identification of 8 new proteins in the apical cap.

### Apical cap and basal structure anchor the helical capsid

The outer rings of the basal structure and the apical cap follow a C14 symmetry and connect the cap complexes with the helical portion of the nucleocapsid, functioning as an anchor. We term this structure

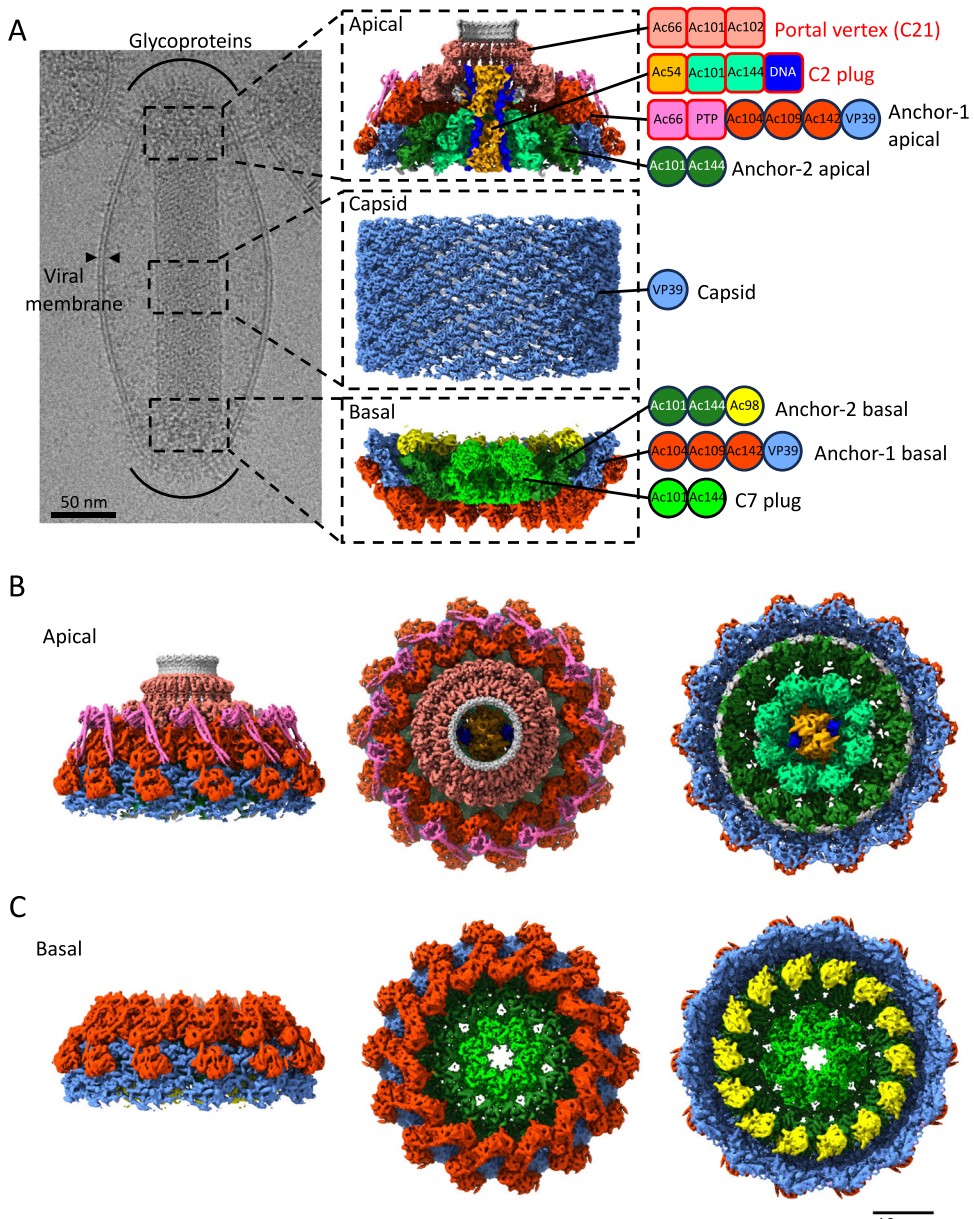

**Fig. 1 | Single particle cryo-EM reconstructions of the BV nucleocapsid.**
**A** Electron micrograph of an AcMNPV virion with major structural components labeled. Enlarged 3D reconstructions of the apical cap, the capsid sheath and the basal structure are shown in adjacent panels. Protein composition of each assembly is indicated by their acronyms, grouped by assembly and sub-assembly. Newly identified proteins from this study are represented as squares; previously known proteins are shown as circles. **B** Isosurface views of the 3D composite map of the apical cap: side view (left), top view (center) and bottom view (right). **C** Isosurface views of the 3D composite map of the basal structure: side view (left), top view (center) and bottom view (right).

the 'anchor' complex. It is organized in a two-ring structure, referred to here as anchor-1 for its most peripheral part and anchor-2 for the inner one (Fig. 1 and Supplementary Fig. 4D–G (basal), Supplementary Fig. 9E, F (apical)). The anchor complex of the apical cap is, in many aspects, very similar but not identical to its basal equivalent. On the similarity, common proteins for both basal and apical anchor complexes include Ac142, Ac109, Ac104, VP39 for anchor-1 and Ac101 (also known as C42 or P40) and Ac144 (E27) for anchor-2[11] (Fig. 1A, Fig. 2A–D and Supplementary Fig. 5A, Supplementary Fig. 10A). Notably, the arrangement of three Ac104 molecules and their interactions with VP39, Ac109 and Ac142 in the apical anchor-1 ring mirrors the equivalent basal arrangement observed in both our BV structures and the recently reported ODV structure[11]. The baffling distinction between the two cap complexes is that the anchor-2 rings, which are very similar to each other, interact with innermost components displaying different cyclic symmetries. In the apical cap, anchor-2 connects to the C2 symmetric part of the portal leading to a C14 to C2 mismatch (Fig. 1B and Supplementary Fig. 9E, F) while in the basal structure, it connects to the C7 plug producing a C14 to C7 symmetry mismatch (Fig. 1C and Supplementary Fig. 4D–G). The apical anchor complex also distinctly diverges from its basal counterpart by lacking Ac98 (38 K) in anchor-2 (Fig. 1; Fig. 2C, D; Fig. 3A, B) but featuring additional proteins, such as PTP and Ac66 in anchor-1 (Fig. 1A, B; Fig. 2A, C). The other important distinction is the presence of the innermost portal complex only in the apical cap, wherein, the portal exhibits clear densities for two short (58 bp) double-stranded (ds) DNA helices (Fig. 1A and Supplementary Fig. 11 B, I) hence indicating DNA packaging and ejection both occur via the BV apex[18]. The anchor-2 ring of the apical cap appears to serve as a communication bridge between the anchor-1 C14 ring and the C2 part of the portal.

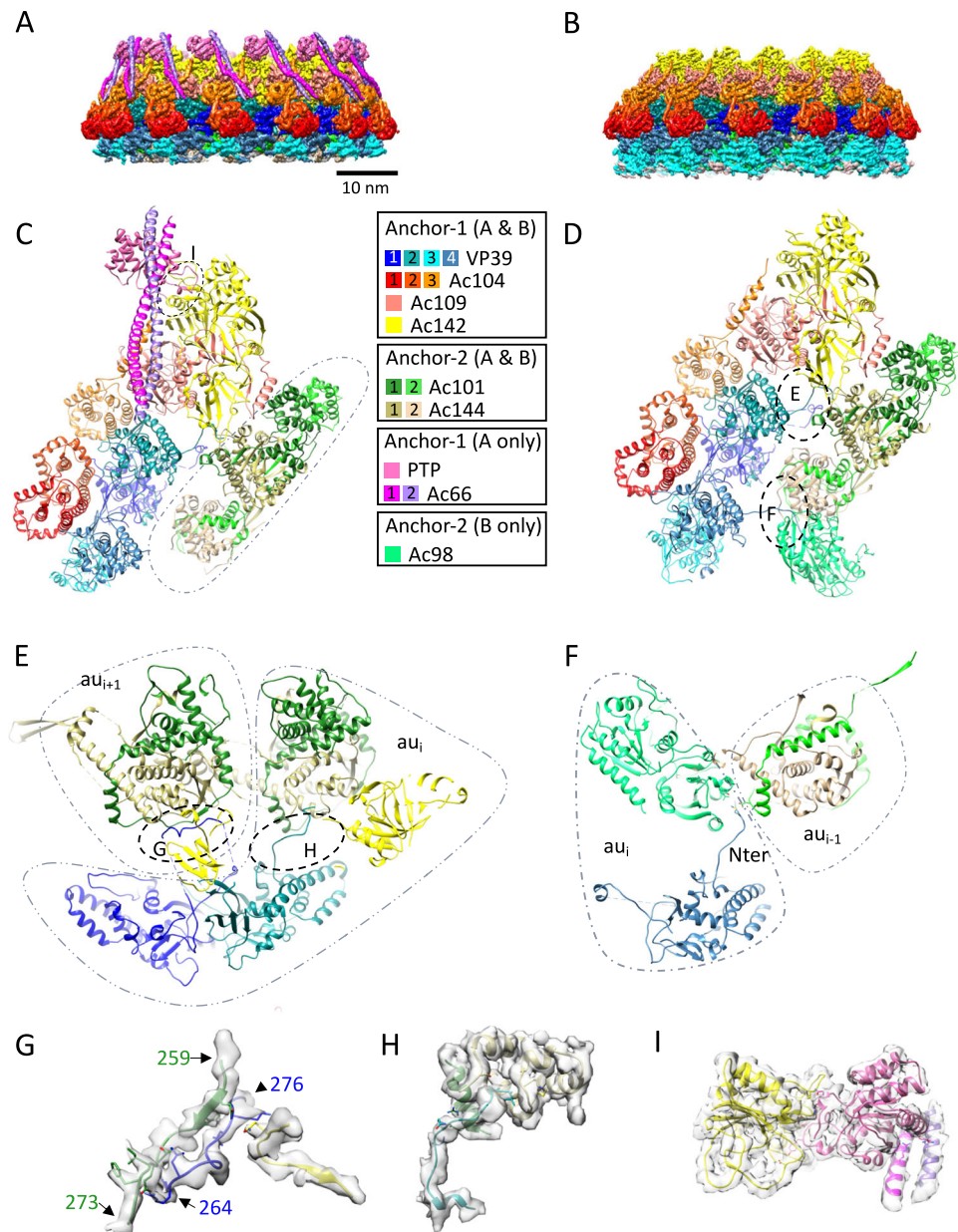

**Fig. 2 | Structures of the C14 anchor complexes from the apical cap and basal structure. A, B** 3D reconstructions of the anchor-1/anchor-2 complex of the apical cap and basal structure, respectively, colored by protein composition (color key below). **C, D** Ribbon representation of the proteins composing the asymmetric unit of the apical and basal anchor-1/anchor-2 respectively colored as in (**A**). The dotted line in (**C**) delineates the anchor-2 complex. **E** Detailed view of the region highlighted in (**D**), showing two consecutive asymmetric units (au$_i$ and au$_{i+1}$). Two key interactions are observed: VP39_1 from anchor-1 of au$_i$ interacts with the Ac144/Ac101 module of au$_{i+1}$. Similarly, VP39_2 from anchor-1 of au$_i$ interacts with the Ac144/Ac101 module of the same asymmetric unit. **F** Detailed view of the region highlighted in (**D**), showing VP39_4 N terminus from anchor-1 of au$_i$ interacting with Ac98 from anchor-2 of au$_i$ and the lower Ac144/Ac101 module of anchor-2 from au$_{i-1}$. Predicted hydrogen bond-forming residues include: M8 for VP39_4, L32 and Y40 for Ac98 and, D273 and N279 for Ac102. **G** Zoomed view of the interaction

region highlighted in (**E**), where residues 264 to 276 of VP39_1 of anchor-1 of au$_i$ engage with residues 259 to 273 of Ac101 and the C-terminus of Ac142 of au$_{i+1}$. Predicted hydrogen bond-forming residues include: N264, R265, L266, L272, K273 for Vp39_1, Q261, Y262, T267, E268, I 269, F271 for Ac101 and Q406 for Ac142. **H** Zoomed view of the interaction region highlighted in (**E**). The N-terminus of vp39_2 of anchor-1 of au$_i$ interacts with both Ac144 and Ac101 of anchor-2 of the same asymmetric unit. Hydrogen bond-forming residues include: L3, Q12, R14 for Vp39_2, E276, R280 for Ac102 and S59, M63, Q65, S200 for Ac144. **I** Close-up of the region highlighted in (**C**) showing the interaction of PTP (pink) with Ac142 (yellow) and an Ac66 dimer (purple and magenta). Predicted hydrogen bond-forming residues are: *PTP/Ac142 interaction:* H7, N8, Y35, T37, E40 for PTP and N180, P209, K225, K216, N218, S220 for Ac142. *PTP/Ac66 interaction:* K109, P111, M113 for PTP, Q315, R322 for Ac66_1 and Q317 for Ac66_2).

## Anchor-1 complex of the apical cap and basal structure

As a whole, the anchor-1 ring of the apical cap comprises the capsid protein, VP39, in conjunction with Ac142, Ac109, Ac104, PTP and Ac66 (Fig. 2C). For both the apical and basal ends, the nucleocapsid's elongated tubular section, composed of helically organized VP39 dimers, terminates with Ac109 binding. Additionally, the VP39 lattice

transitions from a straight to a curved shape due to interaction with proteins forming the anchor-1 and −2 rings. The main influencing factor for the curvature is the presence of three Ac104 copies per asymmetric unit, each interacting with a distinct VP39 monomer (Fig. 2C). Furthermore, VP39 monomers in the cap structures, totaling four per asymmetric unit, exhibit slight variations in conformation

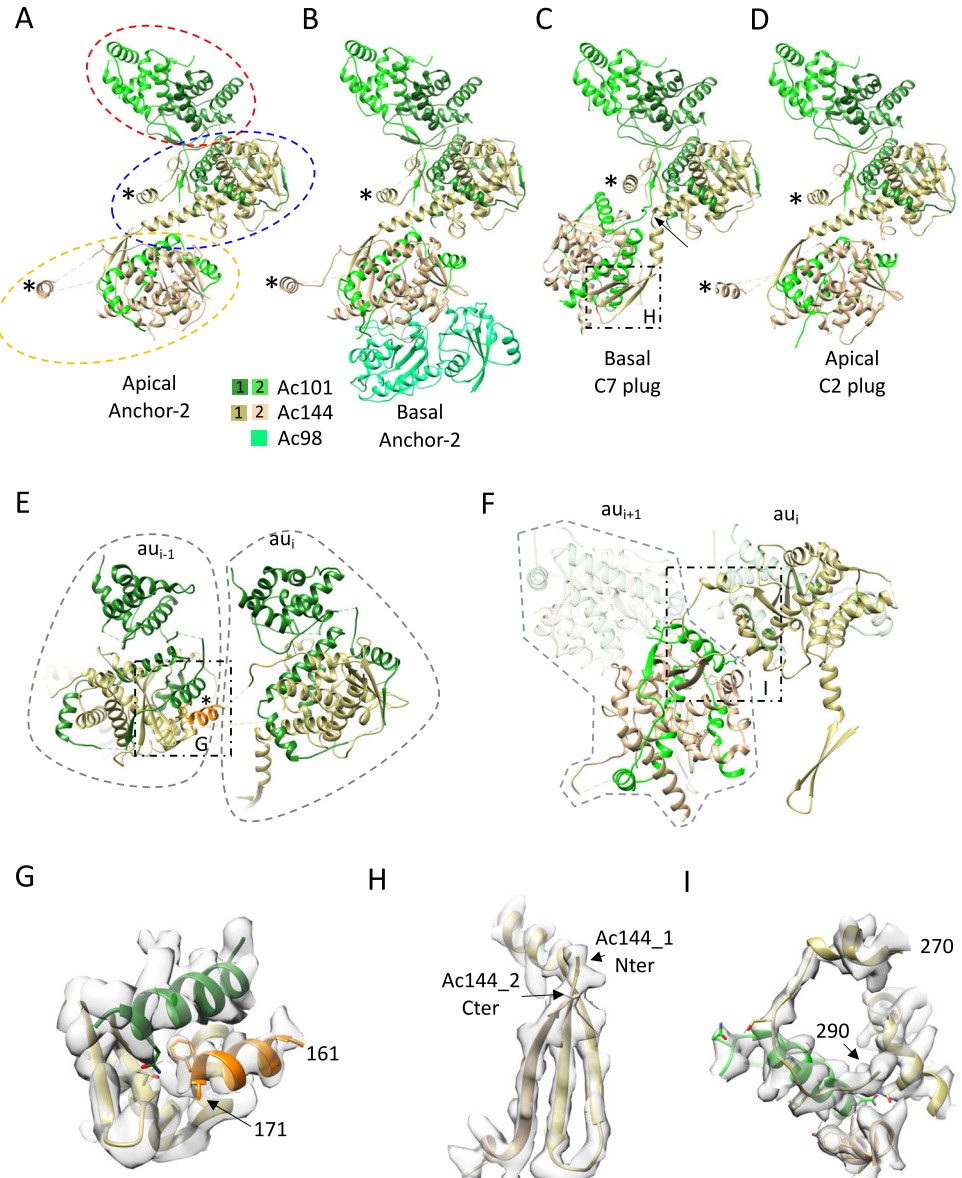

**Fig. 3 | The versatile nature of the Ac101/Ac144 complex is conducive to the formation of different symmetries in the nucleocapsid. A–D** View of the assemblies formed by two hetero dimers of Ac101/Ac144 in the apical and basal anchor-2 complexes (A and B, respectively) as well as in the basal C7 plug (**C**) and the apical C2 plug (**D**). In (**A**), the homo dimer formed by residues 112 to 219 of Ac101 as well as the upper and lower Ac101/Ac144 modules where Ac101 wraps around Ac144 are highlighted by a dotted red, blue and orange oval respectively. The * indicates the position of the α-helix formed by residues 161 to 172 which is involved in inter asymmetric unit interactions between the Ac101/Ac144 modules except for the lower Ac101/Ac144 module of the C7 plug of the basal structure. In (**C**), the arrow indicates a kink in the long α-helix of the upper Ac144_1 which lead to a completely different relative position of the lower Ac101/Ac144 module relative to the upper one in the basal C7 plug. **E** Detailed view of the interaction between asymmetric units in the basal C7 plug (named $au_i$ and $au_{i-1}$) mediated by residues 161 to 172 of Ac144 of $au_i$ (in orange, * in (**E**) and in (**A–D**)). **F** Inter asymmetric unit interaction in the basal C7 plug only. The C-terminus of the upper Ac144_1 of $au_i$ is interacting with the lower Ac101/Ac144 module of $au_{i+1}$. **G** Zoomed view of the region in (**E**). Predicted hydrogen bond-forming residues include: F170 for Ac144_1 of $au_i$, Q255 for Ac101_1 and T126 for Ac144_1 of $au_{i-1}$. **H** Zoomed view on the region highlighted in (**C**). The upper Ac144_1 N-terminus interacts with the lower Ac144_2 C-terminus by β-sheet augmentation. **I** Zoomed view on the region highlighted in (**F**). The upper Ac144_1 C-terminus (residues 270 to 290) of $au_i$ is indicated. Predicted hydrogen bond-forming residues include: N246, S281, S285 for Ac144_1 of $au_i$, Q244, Q258 for Ac101_2 and Y234 for Ac144_2 of $au_{i+1}$ respectively.

compared to those in the helical lattice to accommodate the specific protein environment of the cap complexes. In particular, as shown for ODV, in our BV structure also we observe that the loop encompassing residues 180 to 192 of VP39 is adopting different conformations (Supplementary Fig. 1H, Supplementary Fig. 5C) because of the Ac104 binding[11]. Besides Ac104, the binding of Ac109 to VP39-2 induces a re-organization of the region upstream of the spine helix (residues 163 to 176) (Supplementary Fig. 1I, Supplementary Fig. 5C right panel) which allows the formation of a disulfide bond between Cys187 of Ac109 and Cys169 of VP39 (Supplementary Fig. 5D). Moreover, from our

reconstruction of the basal anchor complex for which a slightly higher resolution (4.1 Å) than the apical one (4.7 Å) has been obtained, few interesting observations are made: 1) It appears that residues 253 to 284 of VP39, which are involved in axial inter-subunit interactions in the helical lattice (Supplementary Fig. 1F, G), are disordered in three of the four VP39 monomers of the basal structure (Fig. 2D, Supplementary Fig. 5C left panel). The same region in the 4th monomer (VP39-1) exhibits a well-defined density and adopts a loop conformation (through residues 264 to 276) that interacts with Ac101 (residues 261–271), Ac142 C-terminal and Ac144 of a neighboring asymmetric

unit (Fig. 2E, G and Supplementary Fig. 5C left panel). 2) The N-terminal residues 1 to 14 of VP39-1 and VP39-3 are disordered as in the helical lattice of the nucleocapsid sheath, while they are ordered in VP39-2 and VP39-4. This new conformation of the N-terminus of VP39-2 enables it to extend away from the VP39 parallelepiped-shaped body to interact with the anchor-2 proteins Ac144/Ac101 (Fig. 2E, H and Supplementary Fig. 5C right panel) while the ordered N-terminus of VP39-4 (from the bottom most VP39 layer of the basal structure) fits in between Ac98 from the same asymmetric unit and Ac101 from a neighbor asymmetric unit (Fig. 2F and Supplementary Fig. 5E). These aforementioned features in VP39, seen in both nucleocapsids ends, therefore indicate that VP39 exhibits an inherent adjustable fold and shows extensions that can adopt different conformations depending on the interacting proteins in the local environment.

### Anchor-1 proteins specific to the apical cap

By comparison with the basal structure, the 3D reconstruction of the anchor-1 ring of the apical cap exhibits marked additional densities on the outer surface composed of a globular domain interacting with Ac142 and two long tubular structures, likely coiled coil helices (Fig. 2A). Because of the limited resolution in that region, the identity of these unknown densities could only be found out by docking of Alphafold models of all the BV proteins using DomainFit[20] algorithm in ChimeraX[21]. This approach identified un-ambiguously PTP as the protein occupying the globular density (Fig. 2C, I) with a correlation score of 0.9395 and a Pvalue of $2.22^{e-16}$. Then, Alphafold was again used to search candidates amongst the whole AcMNPV proteome putatively interacting with PTP. Because of the nature of PTP, as being a protein phosphatase, visual inspection of predicted aligned error (PAE) plots suggested too many interacting partners. Hence, we plotted the interface predicted template modeling (ipTM) and predicted template modeling (pTM) scores from this second run of Alphafold and identified Ac66, a known component of the BV nucleocapsid[1,22] (see below), as a potential partner for interacting with PTP (Supplementary Fig. 10C). The Alphafold model of PTP/Ac66 heterodimer suggested PTP interacts with the tip of a long alpha helix of Ac66, specifically residues 301 to 433. This long helix of Ac66 fitted well into one of the two coiled-coil densities. Ac66 itself, is predicted by Alphafold to form dimers through different domains separated by flexible linkers. Interestingly, one of the dimerization domains is the helix interacting with PTP. A third Alphafold prediction for PTP with a dimer of Ac66 confirmed that hypothesis (Supplementary Fig. 10E, F). This dimer prediction is supported by a dimer of the N-terminal domain of Ac66 (residues 1 to 75) also present elsewhere in the apical cap (see below). Therefore, we propose that Ac66 dimerizes on the surface of the anchor-1 complex of the BV apical cap, forming a coiled coil between residues 301–433 and through which it binds onto the anchor-1 complex with PTP serving as a stabilizing adaptor.

In BVs, PTP is sandwiched between Ac142 and Ac66. PTP interacts with Ac142 through complementary surfaces, primed to potentially establish hydrogen bonds with residues from the loop spanning residues 209–220 in Ac142, as well as two marginally protruding loops (1–10 and 34–41) at its N terminus (Fig. 2I). The interactions of PTP with Ac66 is at the opposite side of the PTP/Ac142 interaction, and involves primarily one of the two coiled-coil helices. The residues Q315 and R322 from the first helix of Ac66 and Q317 from the second one form putative hydrogen bonds with the backbone oxygens of PTP residues located between two helix/strand motifs (residues 74 to 79 and 109 to 113) (Fig. 2I).

### Structure of the basal and apical anchor-2 complexes

The anchor-2 complex in the apical cap follows a C14 symmetry as anchor-1 ring, but is formed by only two proteins, Ac101 and Ac144, each being present in two copies per asymmetric unit. Their structural organization closely resembles that in the basal structure of the current BV structure and the reported ODV structure[11]. Ac101 dimerization involves residues 112 to 219 of each protomer (Fig. 3A). Following an adaptable linker, residues from 243 to the C-terminus (residue 334) of each Ac101 monomer elegantly wrap around the globular shaped domains of each Ac144[11] forming a heterodimer. Ac144 dimerization involves an interaction between the N-terminus from one Ac144 and the C-terminus of the second Ac144 (Fig. 3A, H and Supplementary Fig. 6E) through β-sheet augmentation. This peculiar assembly of the two proteins is facilitated by the long linker between Ac101 N- and C- terminal domains that can adopt different conformations, and being in a more elongated state in one of the two Ac101s. In a side view, the anchor-2 asymmetric unit appears as a sequential assembly of three modules: the dimer of Ac101 followed by the first and second Ac101/Ac144 modules (Fig. 3A).

Anchor-2 asymmetric units interact with each other through the dimeric modules of Ac101. Another key interaction involves a short α-helix, consisting of residues 160 to 170, of Ac144 of one asymmetric unit that docks on the neighbor Ac101/Ac144 module (Fig. 3E, G and Supplementary Fig. 6F). This short α-helix can be in different positions relative to the core of Ac144 depending on which Ac144/Ac101 modules it belongs to in the anchor-2 (Supplementary Fig. 13A) or, in a global view of the nucleocapsid itself, on which cyclic symmetry the Ac144/Ac101 modules belong to (see below). The α-helix is part of an extended loop structure from residues 155 to 200 that protrudes out of the globular part of Ac144. This very modular loop region, including the α-helix, is therefore critical for maintaining interaction between asymmetric units despite different local environments and for engaging the Ac101/Ac144 assembly in different cyclic symmetries (see below).

A major difference between the anchor-2 complexes in the apical and basal structures is the presence of Ac98 in the basal anchor-2, which interacts exclusively with Ac144 from the lowermost Ac101/Ac144 module (Fig. 3B). Nevertheless, this interaction with Ac98 has very little consequence on the structure of Ac144 or on the rest of the anchor-2 (Supplementary Fig. 13B). In the apical anchor-2 3D map, some additional densities are visible at locations that would be Ac98 in the basal structure (grey density in Fig. 1B right panel) but their identity could not be assessed. While it has been claimed that Ac98 could be distributed throughout the sheath and not particularly affiliated to the end structures[23], our BV structure and the published ODV structure[11] have clear densities for Ac98 only in the basal structure.

### Structure of the C7 plug of the basal structure

The inner part of the basal structure has a C7 symmetry and functions as a plug closing the void created by the peripheric C14 anchor complex. The C7 ring, like the anchor-2 complex, is also composed of the two proteins Ac101 and Ac144. However, a major difference in the C7 plug involves one of the Ac144 (Ac144_1) monomers, where a kink is introduced between residues S38 and L39 in the long α-helix (residues 28–60) (Fig. 3C and Supplementary Fig. 13A right panel). This kink results in a large re-orientation of the lower Ac101/Ac144 module (Ac101_2/Ac144_2) leading to the formation of a closed plug structure instead of an open ring as in the anchor-2 C14 ring[11]. Additionally, due to the aforementioned kink, the C7 plug shows distinct inter-subunit interactions. While the inter-subunit interactions involving the dimeric domains of Ac101 (residues 122 to 219) and residues 160-170 of Ac144_1 remain similar to the anchor-2 configuration, the globular cores of the lower Ac101/Ac144 module now face each other directly (Fig. 3F). This organization is stabilized by the C-terminus of the upper Ac144_1 (residues 270-290) (Fig. 3F and Supplementary Fig. 6G). In contrast to the basal or apical anchor-2 complexes, where this region of Ac144 is disordered, it is ordered in the C7 plug allowing it to interact with the lower Ac101/ac144 module of neighboring units (Fig. 3I; Supplementary Fig. 13A left panel). These observations at the C7 plug for Ac101/Ac144 proteins further expand the conformational space this modular assembly can adopt.

### Structure of the C2 plug of the BV portal

The innermost and lowest portion of the apical cap was identified as having C2 symmetry. Together with the C14 symmetry of the peripheral anchor rings, the majority of the apical cap seems to exhibit C2 symmetry (Fig. 1B right panel, Supplementary Fig. 9E, F). As aforementioned, the most stable part of this entire apical assembly is the peripheral C14 anchor complex with a resolution of 4.7 Å while the inner C2 plug of the portal complex was solved at a lower resolution of 6.1 Å.

Like the anchor complex, the C2 plug structure also consists of two sub-assemblies. First and intriguingly, the peripheral part is composed again of Ac101/Ac144 modular assemblies (Fig. 3D), as in the apical and basal anchor-2 rings, and in the C7 plug. In total, it is composed of eight such Ac101/Ac144 modules, of which four are unique. The latter are almost identical to each other with very slight variations (Supplementary Fig. 14A), thus the peripheral part of the C2 plug can be considered as arranged in a pseudo-C8 ring. The Ac101/Ac144 assembly within this C2 (or pseudo-C8) part of the portal adopts a heterodimeric arrangement very similar to that of the anchor-2 complexes of the apical and basal caps (RMSD less than 1 Å) (Fig. 3D). Indeed, Ac144_1 from the upper Ac101/Ac144 module shows a straight long N-terminal α-helix which allows for the formation of an open pseudo-C8 ring similar to the C14 anchor-2 rings.

Analogous to the symmetry mismatch found between the C14 anchor-2 and the C7 plug at the basal structure, a disparity is observed at the apical cap between the C14 anchor-2 and the C2 plug. Interestingly, once again this symmetry mismatch remains compatible and allows cohesive assembly of the different structures. However, unlike a straightforward 2:1 interaction observed for C14:C7 in the basal structure and detailed for ODV[11] and, although a logical 7:4 interactions would be favorable for C14:C2 (or pseudo C8), we observed an intriguing pattern of interactions between each Ac101/Ac144 pair of the C2 ring and their counterparts of the C14 ring. Specifically, the four Ac101/Ac144 pairs encompassed within the C2 ring asymmetric unit, designated C2-1, C2-2, C2-3 and C2-4 (Supplementary Fig. 14A) interact, respectively, with 1 (C14-1), 1 (C14-3), 2 (C14-4 and C14-5) and 2 (C14-6 and C14-7) C14 asymmetric units leaving, at first glance, the C14-2 subunit of the C14 ring with no interactions with the C2 plug (Supplementary Fig. 14B), thus compensating for a C14 to a C2 mismatch and yielding a 6:4 interaction.

This pseudo-C8 ring surrounds an inner sub-structure composed of 2 copies of Ac54 dimers anchoring two dsDNA (Fig. 1B right panel, and Fig. 4A) and together closes the apical opening, thus acting as a plug. A small additional density that corresponds to another protein domain bound to each DNA helix could not be identified herein (Supplementary Fig. 11B and E). The pseudo-atomic structure of the Ac54 protein was generated using Alphafold and confirmed by docking in the EM density (Supplementary Fig. 11J). Ac54 has been previously identified as a structural protein in both BV and ODV[17] and proposed to be indispensable for nucleocapsid assembly[24], transport of capsid proteins to the nucleocapsid assembly site[25] and viral DNA encapsulation[26]. Ac54 is composed of 365 amino acids and the EM density has allowed us to model most of the protein except for the first 6 residues and a part of a disordered loop. Ac54 consists of an N-terminal domain containing mainly β-strands and an α-helical C-terminal domain (Fig. 4B). Interestingly, residues 7 to 40 form an extended loop acting as the primary component in forming the dimer interface (Fig. 4B). Specifically, residues 22 to 25 of one monomer interacts with residues 205 to 207 of the other monomer albeit through only hydrogen bonds. It is noteworthy that a DALI[27] search for potential fold-matching proteins yielded no convincing results with high Z scores, questioning whether Ac54 displays a new fold.

In total, 4 dimers of Ac54 are visible within the C2 plug for anchoring the two dsDNA moieties. Interestingly, each Ac54 dimer binds both dsDNA strands through 'echelon' arrangement along the two DNA helices (Fig. 4B, C). Although protomers 1 and 2, as well as protomers 3 and 4, form strict dimers, each subunit of the dimer shares distinct interactions with the dsDNAs. Specifically, protomer 1 primarily interacts with the dsDNA of its symmetrical counterpart (Fig. 4D, E), while protomer 2 interacts with its own dsDNA (Fig. 4F). This pattern is mirrored in the bottom protomers, where protomer 3 interacts with the dsDNA of its C2 counterpart and protomer 4 interacts with its own (Fig. 4H, I). Several arginine and lysine residues line up the channel to interact with the dsDNA with protomer-1 sharing more interactions with dsDNA compared to other three protomers. While all the four protomers use Arg255 and Arg308 for dsDNA interactions, only protomer-1 also involves N-terminal residues, Arg45 and Arg47. Additionally, Lys298 of protomer-1, Lys307 of protomer-2 and Lys298 and Lys305 of protomer-4 also contribute to the dsDNA interaction. Sequence alignment of Ac54 analogs from 28 other Alphabaculoviruses showed that both the Arg255/Arg308 and Arg45/Arg47 tandems are strictly conserved suggesting a conserved way of DNA binding amongst baculoviruses (Supplementary Fig. 15)[28].

This Ac54-DNA assembly is also supported by a network of protein-protein interactions, intriguingly involving only protomers-2 and −3 (Fig. 4J, K). First, Ac54 protomers −2 and −3 each form a disulfide bond with their corresponding C2 counterparts, via Cys13 residue (Fig. 4G). Then, protomer −2 interacts with two Ac101 subunits, one each of C2-2 and C2-3 asymmetric units from the pseudo-C8 ring (Fig. 4L). Potential interactions involve residues 190-200 of Ac101s and the C-terminal residues of Ac54 (340 to 360). More interactions are observed with protomer-3. It interacts similarly to protomer-2 with two Ac101 subunits of the C2-1 and C2-2 asymmetric units of the pseudo-C8 ring, but also with Ac101_2 and Ac144_2 subunits from the lower part of the C2-3 asymmetric unit. (Fig. 4M). The interacting residues of protomer-3 are distributed throughout the entire protein, with a higher number of interactions involving the N- and C-terminal regions (Fig. 4M). In spite of the different environment involving various interactions with the pseudo-C8 ring and with the dsDNA moieties, the two Ac54 dimers within one asymmetric unit superpose noteworthily onto each other (Supplementary Fig. 11K). This above organization favors the possibility that DNA encapsulation could be efficiently triggered and carried out involving conformational changes between different units of the portal.

### Structure of the C21 vertex of the BV portal

The BV apical cap displays a prominent crown-like structure sitting right above the C2 symmetric part of the BV portal (Fig. 1A, B), adopting a C21 symmetry (Fig. 5A, B) and being the vertex of the apical cap. It has a maximum and minimum diameter of 23 and 10 nm respectively. The asymmetric unit of the C21 ring observes a distinctive 'sickle' shape (Fig. 5C), characterized by a straight vertical 'handle' domain, connected to a 'hook' domain. Thanks to a higher resolution for this ring structure (3.5 Å) (Supplementary Fig. 12B, C), the extensive modeling analysis indicated three different proteins, each one in two copies, contributing to form this crown-like C21 ring. The handle region comprises a dimeric Ac66, spanning N-terminal amino acids 4-75 (Fig. 5C; Supplementary Fig. 12D), with the remaining amino acids of Ac66 conspicuously absent in the C21 ring structure. Our Alphafold prediction for Ac66 dimer (Supplementary Fig. 10E) closely aligns with the N-terminal segment, built ab-initio from Modelangelo[29], reinforcing our confidence in categorizing this segment as part of Ac66. The rest of the Alphafold prediction for Ac66 dimer alternates between unstructured and dimerization domains (Supplementary Fig. 10E). One such dimerization domain, which includes identified residues 310 to 386, forms a coiled coil domain interacting with PTP as part of the apical anchor-1 complex (see above, Fig. 2A, C). We hypothesize that the missing segments of Ac66 dimers (besides N-termini and residues from 310 to 386) are highly flexible and/or averaged out through symmetry considerations, therefore posing challenges in their observation.

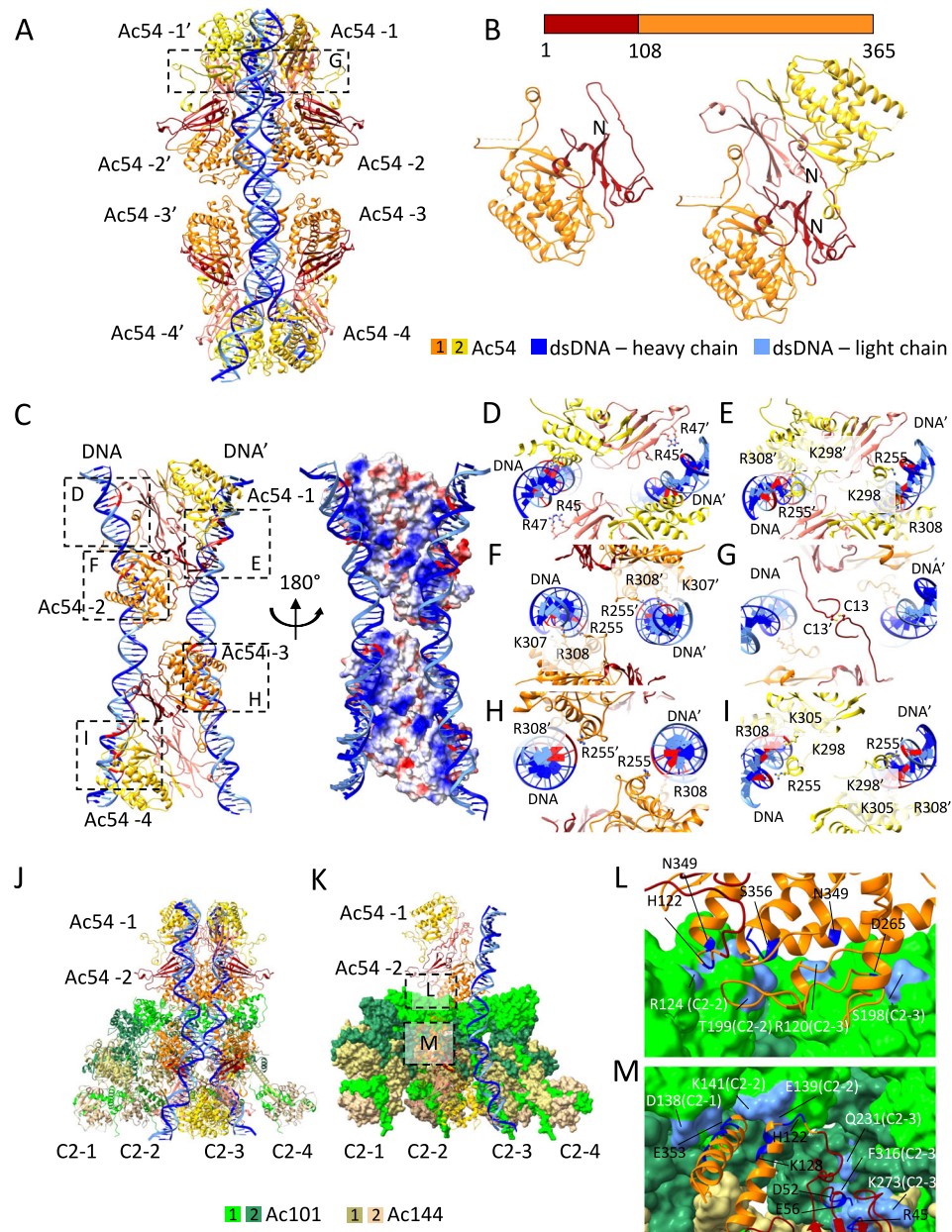

**Fig. 4 | C2 inner plug of the apical cap. A** Ac54 monomers in an 'echelon' arrangement on the dsDNA. Two strands of dsDNA are delineated very well in the structure. Inset (**G**) is marked here. **B** N- and C-terminal domain organization of Ac54 in the sequence (top) and monomeric and dimeric structural organization (bottom). **C** Left: Four protomers of two Ac54 dimers interact with dsDNA, each with unique interactions. Only one asymmetric unit containing four protomers is shown for clarity. The C2 counterpart is not shown here. Insets (**D**–**F**, **H** and **I**) are marked here. They are arranged in the order of seeing the assembly from top to bottom Right: Electrostatic surface of the four Ac54 protomers exhibiting positive charged surface poised to bind to dsDNA. **D**–**I** Ac54-dsDNA and Ac54-Ac54 interactions. Residues involved in interactions are shown in ball-and-stick representation. As the dsDNA is a random sequence, potential interacting positions on the dsDNA are highlighted in red. The dsDNA and Ac54 residues of the symmetrical counterpart are marked with a prime ('). **D** N-terminal domain of protomer-1

(Ac54-1) interactions with dsDNA. **E** C-terminal domain of protomer-1 interactions with dsDNA. **F** Protomer-2 interactions with dsDNA. **G** Di-sulfide bridge involving Cys13 of protomer-3 of the two symmetry-mates. **H** Protomer-3 interactions with dsDNA. **I** Protomer-4 interactions with dsDNA. **J** Ac54 interactions with pseudo-C8 Ac101/Ac144 subunits. The asymmetric unit has 4 Ac101/Ac144, marked as C2-1 to C2-4. **K** Same as (**J**) but with space-filled model for the pseudo-C8 ring to clarify the intricate arrangement allowing only the protomer-2 and 3 to engage in interactions with the pseudo-C8 Ac101/Ac144 subunits. Insets (**L**) and (**M**) are marked here. **L** Protomer-2 interactions with C2-2 and C2-3 units. **M** Protomer-3 interactions with C2-1, C2-2 and C2-3 units. C2-4 unit of the pseudo-C8 ring is not involved in any interactions with Ac54 protomers. In (**L** and **M**), potential interacting residues on Ac54 are highlighted in navy blue and those on Ac101/Ac144 subunits are highlighted in cornflower blue.

The 'hook' portion of the asymmetric unit consists of two sets, each constituted of two proteins: Ac101 and Ac102 (Fig. 5C). The first set, encompassing N-terminal amino acids 29-119 of Ac102 (Supplementary Fig. 12E) and 2-78 of Ac101 (Supplementary Fig. 12F), is positioned closer and perpendicular to the handle part of the assembly

whereas the second set, including amino acids 54-117 of Ac102 (Supplementary Fig. 12G) and 2-63 of Ac101 (Supplementary Fig. 12H), is located at the tip of the hook (Fig. 5C). The structure of Ac102, as Ac54, is reported here for the first time. Ac102 is a small protein of 122 amino acids functioning as an essential protein for nucleocapsid

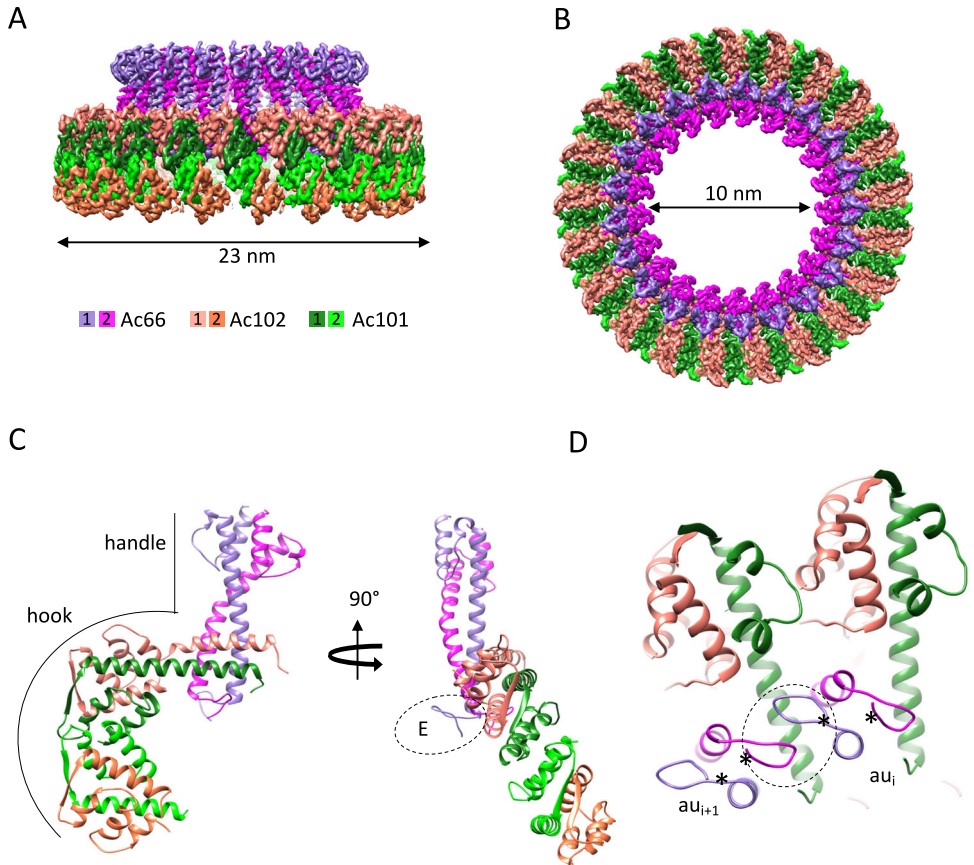

**Fig. 5 | The C21 ring forming the crown of the nucleocapsid. A, B** Isosurface representation (in two 90° related views) of the 3D map of the C21 crown of the apical cap colored by the proteins composing it (color code below panel **A**). **C** Two 90° related views of the proteins composing an asymmetry unit of the C21 crown.

**D** Zoomed view of the interaction at the N-terminal level of Ac66. The N-terminus in each asymmetric unit (labeled *) interacts with each other. The dotted circle highlights the interaction between the N-terminus of different asymmetric units (au$_i$ and au$_{i+1}$).

morphogenesis and implicated in conjunction with Ac101 to play a vital regulatory role for viral nuclear actin polymerization factor activity[30,31]. The protein is primarily an α-helical protein with just one β-strand.

It is noteworthy that almost every protein of the asymmetric unit extensively engages in intermolecular interactions within the C21 ring structure. Within an asymmetric unit, Ac101 pairs up with Ac102 in a heterodimeric arrangement. The interaction between these two proteins is pervasive, extending intimately across the entire modeled length of both proteins, with a spectacular β-strands pairing forming an intermolecular β-sheet (Supplementary Fig. 12I). Almost half of the interactions between Ac101 and Ac102 involves the β-sheet residues in both heterodimers. However, the lower Ac101-Ac102 pair shares more salt-bridges than the upper pair. This is attributed to a different conformation that Ac102 adopts in the lower pair (RMSD between upper and lower Ac102 is 3.5 Å). Ac102 of the first set engages in intermolecular interactions with Ac66 through various polar interactions including a potential salt bridge between the Lys32 of Ac66 with Asp41 of Ac102. The Ac66 arms insert in between two neighboring Ac101/Ac102 sets with the N-terminal part of each Ac66 forming a little hook interacting with residues 72 to 78, notably Gln72 and Tyr73 of Ac101, as well as engaging lateral interactions with other Ac66 neighboring hooks (Fig. 5D).

**Anchoring of the C21 ring onto the C2 core of the apical cap**
In addition to the intricate protein network connections visible within the C21 ring, diffuse densities can be observed in sections witnessing that the C21 ring could be directly anchored to the rest of the C2 symmetric apical cap. Indeed, for some of the proteins assigned

within the apical cap, in particular Ac101 and Ac66, only portions or domains could be identified and found to be present in different subcomponents of the apical cap. Regarding Ac101, while the N-terminal regions (2-78 and 2-63 for Ac101_1 and Ac101_2 respectively) assemble into a three α-helical bundle and interacts with Ac102 and Ac66 in the C21 ring, the rest of Ac101 from residue 112 is an integral member of both the C14 anchor-2 complex and the pseudo-C8 ring. Specifically, in both these two latter structures, residues 112–219 of Ac101 undergo self-dimerization and residues 243–334 wrap around Ac144. The linker residues, 79–111, of Ac101 remain unmodelled and supportively, AlphaFold predicts an extended, entirely unstructured linker from residues 64 to 116 as well. Therefore, this linker is long enough to cover the distances spanning between the currently determined Ac101 domains located in the different apical substructures (between 40 Å and 80 Å). Thus, it is tempting to suggest that Ac101 assumes a pivotal role as a connector between the C21 ring and the C14 anchor-2 complex and the C2 plug of the portal on the other side (Fig. 6A). Intriguingly, in the C21 ring, 42 N-termini of Ac101 are present while there is a combined total of 44 copies of the C-terminus of Ac101 in the anchor-2 (28 copies) and in the C2 plug (16 copies). It appears that two Ac101 N-termini originating from the C14 anchor-2 ring and/or the C2 part of the portal are not incorporated in the C21 ring and are most probably flexible or used for other unidentified purposes.

Regarding Ac66 distribution among different apical rings, its N-terminus (residues 4 to 76) forms a homodimer in the C21 ring accounting for a total of 42 copies of Ac66. This large number of copies of Ac66 agrees with mass spectrometry measurement which identifies Ac66 as one of the most abundant proteins in BV nucleocapsid after

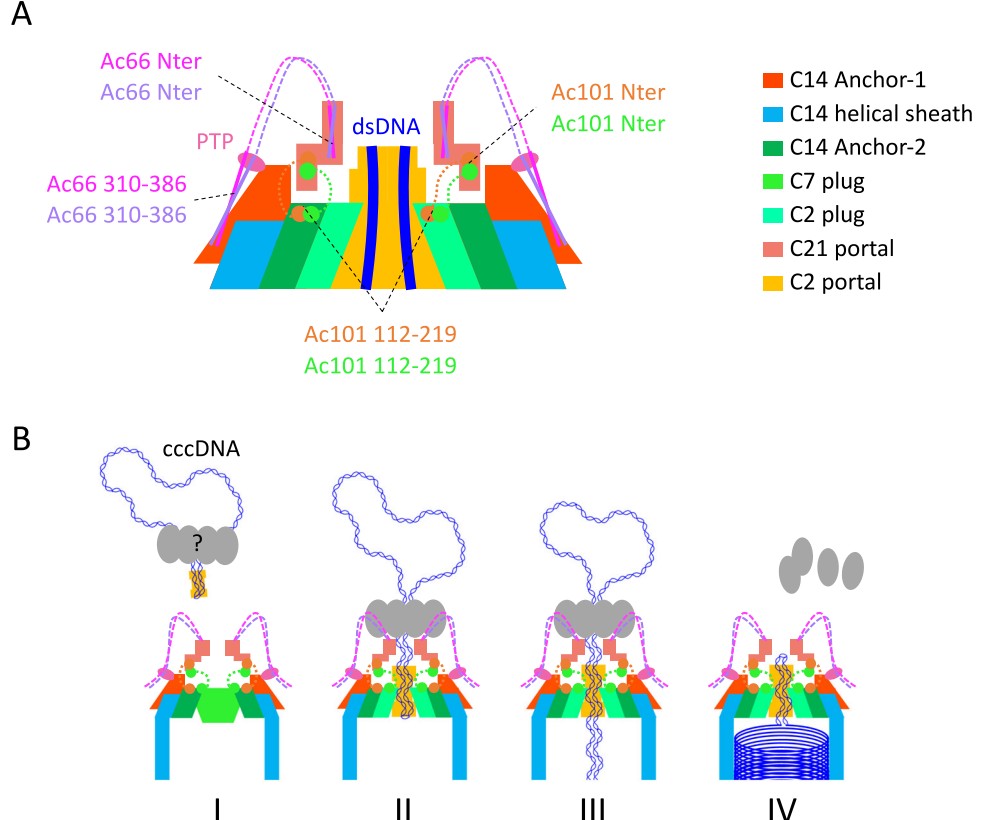

**Fig. 6 | Symmetry mismatches and AcMNPV cccDNA packaging mechanism.**
**A** Schematic representation of the AcMNPV apical cap showing the anchoring of the C21 portal vertex onto the various assemblies forming the rest of the mature virion apex. The C21 ring contains Ac66 N-termini while Ac66 residues 310–386 are visible on the C14 anchor-1 ring bound to PTP. Ac101 also participate in the C21 ring through their N-termini whereas Ac101 residues 112-219 are part of both the C14 anchor-2 ring and the C2 plug. The different complexes are color coded according to Fig. 1, see legend on the right. **B** Proposed AcMNPV genome packaging mechanism. I: Preassembled nucleocapsid with overall C7 apical cap recruits newly amplified covalently closed circular DNA (cccDNA) genome decorated with Ac54 and other unknown subunits of the packaging motor (e.g. GTA). II: Genome

docking triggers the C7 plug to adopt a C2 symmetry probably mediated by Ac54 and Ac101/Ac144 dimer. III: Full packaging apparatus assembles and genome translocation, involving putative helical-to-planar mechanism and ASCE P-Loop ATPase, is favored by symmetry mismatch at the interface between the motor and the rest of the nucleocapsid (C21 vs. C2). IV: Complete genome packaging ends up with packaging of cccDNA closure loop and partial dissociation of the translocating motor as probably being dependent on DNA presence and thus allowing for its recycling. Maintained symmetry mismatch between C14 anchor complexes and C2 plug, in mature virion, may favor later apical capsid dissociation for pressurized DNA release in infected cells.

the capsid protein, VP39, and the glycoprotein, GP64[17,32]. However, with residues 310 to 386 of Ac66 forming a dimeric domain on the surface of the C14 anchor-1 ring to interact with PTP, only 28 copies of Ac66 are accounted for in the current model. The remaining 16 copies of Ac66 can be assumed to be part of the unassigned density of another ring structure observed above the C21 ring (Fig. 1A, B right panel−grey density). Despite the discrepancies in copy number, it is tempting to propose that Ac66 as well participates, with Ac101, in the attachment of the C21 ring to the rest of the apical cap (Fig. 6A). Another indirect evidence of the loose attachment of the C21 ring to the C2 apical core through linkers is the observation that the whole C21 ring can be found in slightly different positions relative to the C2 plug (Supplementary Fig. 11L–N).

## Discussion

AcMNPV has a large dsDNA genome (~135kbp) and therefore most probably requires an energy-dependent mechanism to package its genome, as a prerequisite to give rise to mature DNA-filled virions. Although this process has been studied in dsDNA phages[13], herpesviruses[14,16] and white spot syndrome virus (WSSV)[33] it has never been examined at a molecular level for circular dsDNA virus such as baculoviruses. Several viral proteins have been proposed to be part of the nucleocapsid and take part in the above functions[18,34]. However,

structural evidences have been lacking albeit the recent structural analyses of ODV[10,11] that have yielded valuable insights into the nucleocapsid sheath and basal structure but only a partial view of the apical cap (specifically only the anchor complex). The apical cap structure has been proposed to be responsible for the packaging of genomic DNA[34]. Expanding upon this body of knowledge, our cryo-electron microscopy structure of the BV now provides high-resolution insights into the apical cap involved in DNA packaging and also bridges various structural gaps from the published structures. Despite being widely used since decades as biotechnology tools, we report here for the first time in situ structures of the portal of AcMNPV budded virion. Our structures identify the apical end as being the portal for DNA encapsulation and ejection. From our atomic models, we identified the apical cap consisting of several concentric substructures: four ring-like structures circling two dsDNA moieties pinched by Ac54 dimers. From the outside is found a C14 ring, named apical anchor-1, containing the newly identified PTP and Ac66 proteins and rendering it different from the basal one[11]. Towards the inside, another C14 ring, the anchor-2, is found very similar to the basal one. Inner to it, a C2 plug harboring a pseudo C8 symmetry is visible that surrounds two Ac54-bound dsDNA strands buried inside the apical cap and poking out of the nucleo-capsid sheath. Lastly, a C21 ring sits onto these C2-symmetric sub-assemblies and forms the vertex of the apical cap of the BV

nucleocapsid. The overall molecular composition of the apical cap determined herein matches the phenotypes observed when the identified baculoviral proteins were mutated in previous studies as well as the functions proposed to happen at this pole of the nucleocapsid, such as capsid formation and egress or DNA packaging. Indeed, previously, Ac101 and Ac144 were identified to be essential for proper nucleocapsid formation[35–37]; Ac104 was described being a capsid associated protein[36] essential for nucleocapsid egress from the nucleus[38]; PTP was found to be required for AcMNPV replication[39] and to play a major role, as a virion associated structural protein, in the induced Enhanced Locomotory Activity (ELA) observed in BV-infected hosts[40,41]; Ac66 was proposed to be part of the ODV and BV nucleocapsid[17] and to take part in its assembly and egress[22]; Ac102 was described as a nucleocapsid protein[42] required for viral spread[31] and Ac54 was identified as critical for nucleocapsid assembly[25] and proposed to be involved in DNA packaging[26].

A striking feature of the overall nucleocapsid architecture rests upon the structurally highly versatile Ac101/Ac144 heterodimer. The proteins Ac101 and Ac144 are essential BV genes of AcMNPV[43] and have been postulated to be multi-functional involving DNA processing, genome packaging and nucleocapsid morphogenesis and as well in actin dynamics due to its association with p78/83[44,45]. The multi-functional ability of this pair can be attributed to its wide spreading over the molecular organization of both virion poles. Despite the BV nucleocapsid being constituted of dozens of individual proteins, it clearly appears from our structures that the Ac101/Ac144 module should be considered as the main building block of its architecture. Thanks to its structural versatility, assemblies made of Ac101/Ac144 are able to form rings adopting different symmetries and harboring various dimensions. In our structure, we have observed four different conformations for this heterodimer, one each in both apical and basal C14 ring complexes, and within the basal C7 plug and in the apical C2 (or pseudo-C8) plug (Fig. 3A–D).

The work presented herein discloses another intriguing facet of the BV nucleocapsid: a portal at the apex serving as a gateway for genome packaging and exiting. Indeed, we found a C2-ring consisting of Ac101/Ac144 dimers underneath a C21 ring composed in turn of Ac102 and domains of Ac66 and Ac101 for which the rest of the proteins are found to be components of other apical subassemblies. Buried in the middle of the portal, two Ac54 dimers fasten two dsDNA helices with small unassigned additional densities, likely a domain emanating from proteins found in other apical complexes. Ac54 has been reported to bind purine rich regions of the baculoviral genome[26], which are found to be adjacent to two essential genes (polyhedrin and p78/83) and to also locate at the level of homology regions, themselves proposed to be important for DNA recognition and subsequent packaging into progeny nucleocapsid and serving as origin of replication[46,47]. Our observation of the Ac54/DNA complex being located at the apex of the filled nucleocapsid suggests that it plays a key role in the regulation of genome packaging. We therefore propose such complex to represent a general hallmark of baculovirus correct genome amplification as ensuring presence of critical genes and replication ability. We also postulate that the binding of Ac54 to the preassembled apical cap harboring a hypothetical overall C7 symmetry, triggers the reorganization of the plug from C7 to C2, leading to an opened structure enabling DNA packaging (Fig. 6B). The existence of a global C7 apical cap is supported by i) the copy number of Ac101 present in the C21 ring matching the composition of the anchor-2 ring and a theoretical C7 plug and ii) the peculiar anchoring observed in mature virion of the C2 portal (Supplementary Fig. 14B).

The C21 ring aligns precisely with the C2 plug along the central axis of the nucleocapsid despite appearing to be an independent and dynamic structure in the apical cap. It is configured in a ring structure with varying diameters (Fig. 5A, B). However, at its smallest dimension, the C21 ring is large enough to allow the simultaneous passage of two

double stranded DNA (dsDNA) strongly supporting the theory of covalently closed circular DNA (cccDNA) replication and packaging for baculoviruses as also proposed for other viruses[48] such as herpesviruses[14] and phi-174 phages[49]. Notably, no clearly defined densities bridge from the C21 ring to the main body (C2 symmetric) of the BV apical cap and our results suggest that the interactions occur exclusively through multiple copies of domains of Ac66 and Ac101 located in the different sub-complexes and connected by long, flexible linkers. Ac101, anchoring of the C21 portal vertex onto the C2 plug, could play a role in triggering the DNA ejection: the removal of the C21 crown from the capsid would lead to the dissociation of the C2 plug followed by DNA release. In addition, 3D classification of the portal complex endorses a flexible connection and reveals variations in the positioning of the C21 ring relative to the C2 plug of the apical cap (Supplementary Fig. 11 L-N) showing that the docking of the C21 ring onto the rest of the apical cap is variable with possibly various rotation angles and suggesting possible ratcheting movements. This deliberate "loose" architecture is likely to be essential for the function of the C21 ring. Furthermore, we observed a symmetry mismatch within the apical cap between the C21 ring and the C2 main body (C14 and C2). This structural aspect again suggests that the vertex of the portal, formed by the C21 ring, could play a role in DNA packaging. Indeed, symmetry mismatches prevent a consistent interaction surface reducing the friction between rings and are well known to be found in molecular apparatuses such as flagella or other viral portals[33,50–52]. The coprime character of the C21 and C2 plug to each other allows smooth friction between the two structures. This characteristic is most probably helpful during genome packaging and ejection (Fig. 6A). Lastly, Ac66 resembles kinesin, a well-known motor protein, when submitted to DALI[27], and has been proposed to be involved in DNA packaging[1]. Our observation of Ac66 sitting at the portal vertex and interacting with the phosphatase PTP further support this hypothesis. Indeed, Ac66 and PTP are both reported to be present in BV[17] and ODV[53] preparations and PTP possesses, in addition of an RNA 5'-triphosphatase activity, the ability to hydrolyzes ATP to ADP and GTP to GDP[40,54]. Together as a pair, the motor-like protein Ac66 and PTP are most likely to participate in genome packaging.

Taken together, these hints suggest the C21 ring to be putatively involved as part of the molecular apparatus responsible for DNA packaging and pumping cccDNA viral genome inside the nucleocapsid (Fig. 6B). However, nucleocapsid assembly occur in the nucleus at the level of virogenic stroma where the viral genome is amplified. The BV structure revealed herein depicts circulating mature virion into which genome unit is completely packaged and most probably ready for ejection into the host. Because we report the architecture of mature circulating BV, the interaction between the motor protein and the viral DNA is most likely not visible as genome has already been fully processed into the nucleocapsid. This packaging step is achieved by a translocase complex, most probably remaining at the virogenic stroma to be recycled, for DNA packaging into further nucleocapsids, as for other large dsDNA phages or virus[13]. The screening of the 155 AcMNPV proteins using the conserved domain database (CDD)[55] pinpoints the Probable Global Transactivator (GTA) as the only protein belonging to the Additional Strand Conserved Glutamate (ASCE) superfamily of P-Loop NTPases. Viral ASCE ATPase ring motors are responsible for dsDNA packaging into preassembled capsid, as shown in herpesvirus[56] and bacteriophages[57–59], suggesting GTA could take part and play a key role in the AcMNPV genome packaging. However, such large motor assemblies are dependent on DNA presence and are recycled. Accordingly, this molecular motor is therefore not present onto BV and the C21 ring should then be considered to solely be a component of such apparatus.

In light of our results, we hypothesize a genome packaging mechanism for AcMNPV with milestones involving symmetry mismatches (Fig. 6B). Newly amplified genome decorated with Ac54 and

probably other subunits of the packaging motor, such as GTA, docks onto the apical cap of the preassembled empty nucleocapsid yielding to symmetry change from C7 to C2 and to the opening of the portal. The full packaging apparatus would assemble, involving Ac54, Ac66, PTP and GTA, and starts cccDNA translocation, favored by the C21/C2 symmetry mismatch at the interface of the motor and nucleo-capsid, maybe through a putative helical-to-planar mechanism simi-larly to viral motor containing ASCE P-Loop ATPase[57,60]. Complete genome packaging ends up with the passage of cccDNA closure loop through the vertex leading to the packaging apparatus partial dis-sociation as being dependent on DNA presence and thus allowing for recycling. Conserved symmetry mismatch between C14 anchor com-plexes and C2 plug, in mature virion, may later favor apical complex dissociation for the release of pressurized DNA in infected cells. Although the exact molecular mechanism by which the circular dsDNA is incorporated inside the nucleocapsid through the BV portal remains to be determined, the 3D architecture provided here lays foundations for future work required for more detailed understanding of genome packaging in baculoviruses. In addition, our structure underlines the versatile facet of Ac101/Ac144 assemblies, capable of adopting various symmetries, and supports symmetry mismatch as being present in molecular motors found in nature. Such results can probably benefit protein engineering and design. Lastly, getting more knowledge about BV architecture composition and assembly is of importance to con-tinue engineering and bringing to another level such existing bio-technology tools (BEVS, biopesticide) and for limiting its action as an important factor of insect extinction in many ecosystems.

## Methods

### Baculovirus amplification and virion isolation

AcMNPV baculovirus bacmid was isolated from DH10-EMBacY cells[61]. Bacteria were streaked for blue-white screening onto LB/Agar plates containing Gentamycin, Kanamycin, Tetracyclin, IPTG and Bluo-Gal. White colonies were picked after 24 hours and grown overnight in 3 mL of LB supplemented with Gentamycin/Kanamycin to extract bacmid DNA through alkaline lysis/iso-propanol precipitation as previously described[61,62]. For transfec-tion of isolated bacmid in insect cells, $1 \times 10^6$ Sf21 (Thermo Fisher, cat. num. 11497013) cells/well were seeded on multi-6 well plates in 3 ml of Sf-900 II media. 10 μl of purified bacmid were resus-pended in 200 μl Sf-900 II media with 10 μl TransIT-Insect Transfection Reagent (MirusBio) and incubated at room tem-perature for 15 min. The entire transfection mix was added dropwise to a single well and cells were incubated at 27 °C. V0 viral stocks were harvested collecting the supernatant of trans-fected cells 72 hours post transfection. 1 ml of V0 viral stock was added to 25 mL of fresh Sf21 cells at $0.8 \times 10^6$ cells/mL. Cells were cultured in 250 mL Erlenmeyer flasks while shaking at 27 °C and counted every day to monitor cell proliferation as successfully infected cells displayed arrested proliferation. V1 viral harvest were collected 2 days after proliferation arrest (DPA + 24 h) by centrifugation at $1'000 \times g$. 1.25 mL of V1 viral stocks was added to 250 ml of fresh Sf21 cells at $0.8 \times 10^6$ cells/ml in 2.5 L Erlenmeyer flasks and cells were cultured at 27 °C on a shaking incubator until DPA + 72 h. V2 viral harvests were collected by centrifugation at $4500 \times g$, and concentrated 50 times by high-speed centrifuga-tion at $15'000 \times g$, followed by gentle resuspension in DPBS supplemented with 5% glycerol prior storage at −80 °C.

### Sample preparation for Cryo-EM and image acquisition

3.5 μL of sample were applied to 2/1 Quantifoil holey carbon grids (Quantifoil Micro Tools GmbH, Germany) or 1.2/1.3 Ultrafoil holey gold grids (Quantifoil Micro Tools GmbH, Germany) and plunged frozen in liquid ethane with a Vitrobot Mark IV (Thermo Fisher Scientific) (6 to 7 s blot time, blot force 0). The sample was observed at the beamline

CM01 of the ESRF (Grenoble, France)[63] with a Titan Krios G3 (Thermo Fischer Scientific) at 300 kV equipped with an energy filter (Bio-quantum LS/967, Gatan Inc, USA) (slit width of 20 eV). 41,386 movies were recorded automatically on 3 different grids with a K3 direct detector (Gatan Inc., USA) in CDS mode with EPU (Thermo Fischer Scientific). Movies were recorded in super resolution mode for a total exposure time of 6 s with 30 frames per movie and a total dose of ~30 e⁻/Å². The magnification was ×64,000 (0.675 Å/pixel at the camera level). The defocus of the images varied between −1.0 and −2.2 μm.

### Pre-processing

It should be noted that the published results obtained on the baculo-virus helical nucleocapsid[10] and its basal/apical caps[11] were not known at the moment the image analysis of our dataset was done. That's the reason the information from these prior studies was not used during image analysis.

The movies were first drift-corrected and binned 2 times by Fourier cropping with motioncor2[64]. The remaining image processing was done in RELION 4.0[65]. CTF estimation was done with GCTF[66]. Micrographs with a defocus of less than −0.4 μm or a resolution in CTF more than 6 Å were removed.

### Identification of nucleocapsid apical and basal ends (Supplementary Fig. 2 and Fig. 3)

Using the helical picking mode in RELION, the two ends of the nucleocapsids were manually picked from a subset (1064) of micrographs. Particles were binned two times and extracted in boxes of 250 pixels, overlapping by 85%. A first 2D classification was performed which identified 2D classes for the helical part of the nucleocapsid (see below for the section on the image analysis of the helical nucleocapsid) and for both the apical and basal caps. The 2D class averages from the nucleocapsid's caps were then used to perform a template-based picking on the same subset of micrographs. Following another 2D classification, the 2D classes from the basal and apical ends of the nucleocapsid could be identified. The subset of micrographs was then increased (to 5759) and a template-based picking using the latter 2D classes of the basal and apical caps was used. 2D classification was again used to separate the two types of caps resulting in 4870 and 1276 particles for the basal and apical ends, respectively. The 2D classes of the basal structure were then used to generate a rough 3D initial model with EMAN2[67]. That model, low-pass fil-tered to 40 Å, was used to resolve the main symmetry of the basal structure. Various symmetries ranging from C10 to C20 were tested and only the C14 symmetry gave a 3D map of the basal structure with clear secondary structure details (resolution of 7.4 Å) in the anchor-1/ anchor-2 complexes. The same C14 symmetry was also applied to the apical cap (using the basal structure 3D map as initial model, low pass filtered to 40 Å) and from this, a first 3D reconstruction from the apical cap with C14 symmetry was obtained at 9.1 Å resolution. Then, the basal and apical particles were merged and used to train and pick the entire data set (37453 micrographs) with TOPAZ[68]. Following several rounds of 2D classification, 41464 and 22763 particles were obtained for the basal and apical caps.

### Image analysis of the basal structure

Determination of the symmetry of the central plug of the basal struc-ture and overall 3D reconstruction with C7 symmetry (Supplementary Fig. 3): From the basal 3D map obtained with C14 symmetry (see pre-vious section), it was clear that the center of the 3D map was occupied by a plug structure having the wrong symmetry imposed. A 3D reconstruction with C14 symmetry was obtained for the overall basal structure which was then used to define 3 masks using SEGGER[69] in CHIMERA[70]: the first one encompassing the anchor-1/anchor-2

complexes (having C14 symmetry), the second one around the central plug having a different symmetry from C14 and the third one containing everything else. A multibody refinement[71] was then performed in RELION in conjunction with particle subtraction on the 3 bodies. 2D classification was then performed on the subtracted particles corresponding to the plug of the basal structure leading to 15890 particles. Using as an initial 3D model the C14 3D map of the basal plug low pass filtered to 60 Å, various cyclic symmetries (from C2 to C13) were tested by running different refinement jobs. The refinement with C7 clearly stand out from the other symmetries by converging to a 3D map of 4.0 Å resolution. Moreover, clear α-helices and β-strands could be identified.

Knowing that the basal structure was composed of two sub-complexes of matching symmetries: C7 and C14, a consensus 3D map of the entire basal structure with C7 symmetry was then obtained at 5.2 Å resolution and focused refinements were then done on the C7 plug and the C14 anchor-1/anchor2 complexes (see below).

Focused 3D map of the C7 plug (Supplementary Fig. 3): From the consensus 3D map of the basal structure with C7 symmetry, another multibody job (using the same bodies as above) combined with particle subtraction was done. The subtracted particles corresponding to the C7 plug were then iteratively subjected to 3 cycles of 3D refinements and 3D classification (in 3 classes, no alignment) to select only the best particles. Following CTF refinements, a final 3D reconstruction for the C7 plug of the basal structure was obtained at 3.4 Å resolution (at FSC = 0.143) from 17 468 particles.

Focused 3D map of the C14 anchor-1/anchor-2 complexes of the basal structure (Supplementary Fig. 3): From the multibody job of the consensus 3D map of the basal structure (see above '3D map of the C7 plug'), subtracted particles corresponding only to the anchor-1/anchor-2 part of the basal structure were obtained. Several attempts were made to calculate the best possible C14 3D map of the anchor-1/anchor-2 region but the best result was obtained by starting with the same subset (17468p) of particles that gave the final 3D reconstruction of the C7 plug (see above paragraph). CTF refinement (on the defocus per particle) was then performed, followed by a 3D classification (3 classes, no alignment). In the end, a final 3D reconstruction (imposing C14 symmetry) of the anchor-1/anchor-2 of the basal structure was obtained at 4.1 Å (at FSC = 0.143) from 12,164 particles.

## Image analysis of the apical cap

Determination of the structure of the C21 ring (or crown) (Supplementary Fig. 7): From the apical cap 3D map obtained with C14 symmetry (see section above 'Identification of nucleocapsid apical and basal ends'), it was clear that the central part of the 3D map was occupied by a plug (see next paragraph) and a ring (or crown-like) structures having the wrong symmetry imposed. A 3D reconstruction with C14 symmetry was obtained for the overall apical cap which was then used to define 3 masks using SEGGER in CHIMERA: the first one encompassing the anchor-1/anchor-2 complexes (having C14 symmetry), the second one around the central region having a different symmetry from C14 and the third one containing everything else. A multibody refinement was then performed in RELION. Particle subtraction was then used to keep the signal corresponding only to the crown-like ring of the apical cap. 2D classification was then performed on the subtracted particles to remove outliers and confirm the subtraction went well (Supplementary Fig. 12A). A total of 21190 particles was obtained. Using as an initial 3Dmodel the C14 3D map of the crown-like ring low pass filtered to 40 Å, various cyclic symmetries (from C2 to C30) were tested by running different refinement jobs. The refinement with C21 clearly stood out from the other symmetries by converging to a 3D map of 4.2 Å resolution with well-resolved features such as α-helices. 3D classification (3 classes, no alignment) further identified a subset of 6390 particles which, after CTF refinement, gave a final 3D reconstruction at a global resolution of 3.5 Å (at FSC = 0.143).

Determination of the C2 symmetry of the central plug of the apical cap (Supplementary Fig. 7): After the same multibody job used to determine the structure of the C21 ring of the apical cap (see previous section), particle subtraction was then used to keep the signal corresponding only to the central plug of the apical cap (positioned beneath the C21 ring). 2D classification was then performed on the subtracted particles to remove outliers and confirm the subtraction went well. A total of 12981 particles was obtained. Using as an initial 3D model the C14 map of the plug (low pass filtered to 40 Å), a 3D classification (3 classes, C1, with alignment) gave one class (with 93% of the particles) having features with 2-fold symmetry. A 3D refinement with C2 symmetry, using the latter class as initial model (low pass filtered to 40 Å), gave a first 3D reconstruction of the central plug of the apical cap at 8.3 Å (without any mask applied). The densities for the two dsDNA strands were clearly visible and the atomic models of Ac101 and Ac144 could be fitted with confidence confirming that the proper symmetry (C2) have been applied.

Overall 3D reconstruction of the C2 apical cap core (without the C21 ring) (Supplementary Fig. 7): The apical cap can be divided in 3 sub-complexes of different cyclic symmetries: the anchor-1/anchor-2 (C14), the central plug (C2) and the crown -like portal (C21). Therefore, a symmetry-mismatch is present between the crown-like portal and the rest of the apical cap (the core). In order to obtain a C2 symmetric map of the apical cap core made of the anchor complexes and the plug (matching C2 to C14 symmetries), a two steps approach was used. First, a 3D map of only the central plug with the anchor-2 is obtained (using the corresponding subtracted particles). And second, the anchor-1 is added using subtracted particles having all sub-complexes (only the VP39 helical lattice and the dsDNA genome are subtracted). For each of the steps, a 3D refinement is performed using, as initial model, a composite 3D map (low pass filtered to 10 Å) having the corresponding sub-complexes (central plug + anchor-2 for the 1st step, central plug + anchor-1 and −2 for the 2nd step). This approach ensured that the 3D refinement was not driven by the high mass and symmetry (C14) of the anchor complexes and resulted in a correct structure of the central plug. For the 2nd step, a 3D mask excluding the C21 crown-like portal was used. Finally, a 3D structure with C2 symmetry for the apical cap core (anchor complexes and the central plug) was obtained at 6 Å resolution (at FSC = 0.143) from 26 008 particles.

Focused 3D structure of the entire portal of the apical cap (C2 plug + C21 ring) (Supplementary Fig. 8): Starting from the C2 3D map of the apical cap core (see previous paragraph), a multibody refinement was performed on the entire portal (C2 plug + C21 ring) and the anchor complex. Subtracted particles containing the central plug (C2 symmetry) and the crown-like ring (C21 symmetry) were calculated. A 2D classification was then done on the subtracted particles to remove outliers and confirm the subtraction went well. A 3D refinement followed by a first 3D classification (C1, 3 classes, with alignment) and another 3D refinement gave a consensus 3D map for the overall portal of the apical cap. A final 3D classification (C1, 3 classes, with alignment) allowed the separation of two different populations of particles having a slightly different position of the C21 ring relative to the C2 plug. Class 1 and 2 were reconstructed both to 9.6 Å (at FSC = 0.143) from 8377 and 8357 particles respectively.

Focused 3D map of the central plug of the apical cap: Starting from the consensus 3D map for the overall portal of the apical cap, a multibody refinement focusing this time on separating the C2 plug from the C21 ring was performed. Subtracted particles containing only the central plug (C2 symmetry) were calculated. A 2D classification was then done on the subtracted particles to remove outliers and confirm the subtraction went well. From the 12 477 selected particles, a consensus 3D map for the C2 plug was then obtained followed by a final multibody refinement focusing on the Ac54/dsDNA complex and the surrounding eight Ac144/Ac101 modules. That approach gave two 3D maps (Ac54/dsDNA and Ac144/Ac101 parts with overall resolution of

6.1 and 6.5 Å, respectively, at FSC = 0.143) for the C2 plug with the 'best' resolved features. A composite 3D map for the complete C2 plug was then computed in CHIMERA.

Focused 3D map of the C14 anchor-1/anchor-2 complexes of the apical cap (Supplementary Fig. 8): Starting from the C2 3D map of the apical cap core (see paragraph 'Overall 3D reconstruction of the C2 apical cap core (without the C21 ring)'), a multibody refinement was performed on the entire portal (C2 plug + C21 ring) and the anchor complex (see beginning of paragraph 'Focused 3D structure of the entire portal of the apical cap (C2 plug + C21 ring'). Subtracted particles containing only the anchor complex (C14 symmetry) were calculated. A 2D classification was then done on the subtracted particles to remove outliers and confirm the subtraction went well. Following two iterations of 3D refinements (with C14 symmetry applied) and 3D classifications (3 classes, no alignment), two different types of anchor complexes having slightly different curvatures were isolated. Both lead to 3D reconstructions having resolutions of 4.8 and 4.7 Å (at FSC = 0.143) from 5517 and 8327 particles, respectively. Protein modeling was performed on the 3D map determined at 4.7 Å.

### Image analysis of the helical lattice formed by VP39 (Supplementary Fig. 2)

From the manual picking of 1064 micrographs (see paragraph 'Identification of nucleocapsid apical and basal ends'), 2D classes corresponding the straight, helical part of the nucleocapsid were obtained. The subset of micrographs was increased to 7593 and a template-based picking (using the latter 2D classes) was done. The particles were binned to 2 Å/pixel and classified in 2D first. A featureless tube was used as an initial 3D model to do a first 3D refinement with C14 symmetry (determined from the basal and apical structures, see above). A 3D mask excluding the dsDNA genome was then used to obtain a first 3D map of the helical nucleocapsid where an Alphafold[19] model of a VP39 dimer could be un-ambiguously fitted. From the fitted dimers, initial helical parameters were determined ($z = 42.6$ Å, $\Phi = -7°$) in Chimera. Another 3D refinement was then done which gave a 3D reconstruction at 4 Å resolution. Two independent datasets of 15,751 and 19,459 micrographs were then processed in parallel. For each, the following steps were performed to isolate the best subset of particles: template based autopicking, 2D classification, 2 rounds of 3D refinement and 3D classification (5 and 3 classes, no alignment), CTF refinements (anisotropic magnification, defocus per particle and beam tilt), Bayesian polishing and CTF refinement on the defocus per particle. The best particles from both datasets were then combined, a last 3D classification (5 classes, no alignment) was performed and a final 3D reconstruction was obtained at 3 Å resolution (at FSC = 0.143) from 138,225 particles with the final helical parameters of $z = 42.57$ Å and $\Phi = -7,11°$.

### Proteins composing the apical cap and basal structure

Identification from 3D maps with resolution better than 4Å: For individual reconstructions exhibiting resolution better or close to 4 Å, the asymmetric unit density was subjected to Modelangelo[29] for automatic model building without the input of any primary sequence. This first step allowed the tracing of the most probable amino acids to occupy the un-assigned densities, whose final output sequence was then used to perform a BLAST[72,73] against AcMNPV genome to identify the potential protein. This sequence was then input to FindMySequence program[74] to thread the actual protein sequence onto the cryo-EM density map. Proteins identified this way are Ac66 (C21 ring), Ac101 (C21 ring), Ac102 (C21 ring), Ac101 (C7 plug and anchor-2 of basal structure), Ac144 (C7 plug and anchor-2 of basal cap structure), Ac98 (anchor-2 of basal cap structure).

Identification from 3D maps with resolution worse than 4Å: For cryo-EM reconstructions of resolution worse than 4 Å where

Modelangelo could not be built reliably, we performed a rigid body fitting of Alphafold models predicted for all 155 proteins of AcMNPV in order to identify the right protein. Once an Alphafold model was identified as a likely candidate, Alphafold was run again to identify potential dimeric partners which were then validated by rigid body fitting. Proteins identified this way are Ac142 (anchor-1 of basal structure), Ac109 (anchor-1 of basal structure), Ac104 (anchor-1 of basal structure), PTP and Ac66 (anchor-1 of apical cap), Ac54 (C2 plug).

Model refinement: The atomic coordinates were refined with Rosetta[75] and Phenix[76]. The refined atomic models were visually checked and adjusted (if necessary) in Coot[77]. The final model was validated with Molprobity[78]. The figures were prepared with Chimera and ChimeraX[21,70]. The data collection and atomic model statistics are summarized in Supplementary Table 1.

### Reporting summary

Further information on research design is available in the Nature Portfolio Reporting Summary linked to this article.

## Data availability

Coordinates and cryo-electron microscopy maps have been deposited in the Protein Data Bank and the Electron Microscopy Data Bank under the following accession codes: helical capsid (PDB 9H1S, EMD-51771, complete basal cap – C7 symmetry (PDB 9H2A, EMD-51791), basal cap – C14 anchor complex only (PDB 9H2B, EMD-51792 [https://www.ebi.ac.uk/emdb/EMD-51792]), basal cap – C7 plug only (PDB 9H2C, EMD-51793 [https://www.ebi.ac.uk/emdb/EMD-51793]), apical cap core – C2 symmetry (EMD-51794 [https://www.ebi.ac.uk/emdb/EMD-51794]), apical cap – C14 anchor complex only (PDB 9H2J, EMD-51808 [https://www.ebi.ac.uk/emdb/EMD-51808]), apical cap – C21 ring only (PDB 9H2K, EMD-51809 [https://www.ebi.ac.uk/emdb/EMD-51809]), apical cap – C2 plug composite map only (PDB 9H2H, EMD-51803 [https://www.ebi.ac.uk/emdb/EMD-51803]), apical cap – C2 plug consensus map (EMD-51804 [https://www.ebi.ac.uk/emdb/EMD-51804]), apical cap – C2 plug focused map 1 (EMD-51805 [https://www.ebi.ac.uk/emdb/EMD-51805]), apical cap – C2 plug focused map 2 (EMD-51806 [https://www.ebi.ac.uk/emdb/EMD-51808]). Source data are provided with this paper.

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

## Acknowledgements

The authors acknowledge the Eukaryotic Expression Facility at EMBL Grenoble. MP acknowledges staff of the EM Facility at EMBL Grenoble and B. Arragain for discussions and comments throughout the course of the project. The authors acknowledge the European Synchrotron Radiation Facility (ESRF, Grenoble) for provision of beam time on CM01.

## Author contributions

M.P. generated and isolated samples. G.E. performed cryoEM data collection, image processing and analysis. G.E. and E.K. built and refined structural models with input from MP. M.P. conceived the project and run Alphafold prediction screens. G.E., E.K and M.P. analyzed the data and wrote the manuscript.

## Funding

## Competing interests

The authors declare no competing interests.
