## [Transparent Peer Review file · Nature Communications]

Structure of AcMNPV nucleocapsid reveals DNA portal organization and packaging apparatus of circular dsDNA baculovirus.

Corresponding Author: Dr Martin Pelosse

Version 0:

Reviewer comments:

Reviewer #1

(Remarks to the Author)

The manuscript "Structure of AcMNPV nucleocapsid reveals DNA portal organization and packaging apparatus of circular dsDNA baculovirus" by Effantin et al describes the first structure of a baculovirus nucleocapsid as determined by cryoEM. While structural information on the helical sheath and basal structure is available, there is no published structure of the crucial apical cap that houses the genome packaging apparatus. In this study, the authors determined the complete structure of a baculovirus, allowing them to identify new components in its C14 ring and characterizing the apical cap symmetry mismatch as well as the structure of the dsDNA portal protein. The structure of the DNA portal of AcMNPV is the first one described to date amongst the whole baculoviridae taxon and also for circular double-stranded DNA viruses. The manuscript provides a clear and comprehensive description of the structure of a baculovirus. To this reviewer, the highlights of the results are the structural versatility of the Ac101/Ac144 pair and the elucidation of the baculovirus portal, especially the observed two stretches of dsDNA. It is truly a remarkable structure, and should be of great interest to Nat Comms readers.

Question:

The portal structure is fascinating, and its superficial resemblance to phage portals is remarkable. It seems that it is almost certainly the portal DNA entrance and egress. However, I was not sure what the authors were suggesting regarding the mechanism of packaging. On the one hand, they speculate (probably correctly) that the virus codes for a packaging motor-type protein that only transiently docks to the portal to pump DNA into the shell, and thus is not present in the mature virus. On the other hand they note the resemblance of A66 to the motor protein kinesin and its interaction with the phosphatase PTP, seemingly to suggest that somehow the PTP's ability to hydrolyze triphosphates provides energy to drive packaging? How do the authors envision this working, and how does it relate to the proposed transiently assembled motor? ATPase motors have distinct features easily recognized via sequence. Does the virus code for any common motor proteins (e.g. P-loop ring ATPases??)

Minor issues:

For readers unfamiliar with baculovirus, it would help to have a schematic figure in the introduction showing the overall arrangement of the different structural components (apical, basal, etc), including which parts have known structures.

Line 71 should either read "In contrast, small viruses for which virions assembly often rely on..." or, "In contrast to small to small viruses for which assembly relies on..."

In the discussion, the authors might consider referencing more recent and comprehensive models of dsDNA packaging such as:

1) Pajak J, Prokhorov NS, Jardine PJ, Morais MC. The mechano-chemistry of a viral genome packaging motor. *Curr Opin Struct Biol.* 2024 Dec;89:102945. doi: 10.1016/j.sbi.2024.102945. Epub 2024 Nov 4. PMID: 39500074.

2) Woodson M, Pajak J, Mahler BP, Zhao W, Zhang W, Arya G, White MA, Jardine PJ, Morais MC. A viral genome

packaging motor transitions between cyclic and helical symmetry to translocate dsDNA. *Sci Adv.* 2021 May 7;7(19):eabc1955. doi: 10.1126/sciadv.abc1955. PMID: 33962953; PMCID: PMC8104870.

3) Pajak J, Dill E, Reyes-Aldrete E, White MA, Kelch BA, Jardine PJ, Arya G, Morais MC. Atomistic basis of force generation, translocation, and coordination in a viral genome packaging motor. *Nucleic Acids Res.* 2021 Jun 21;49(11):6474-6488. doi: 10.1093/nar/gkab372. PMID: 34050764; PMCID: PMC8216284.

Reviewer #2

(Remarks to the Author)

The authors studied the structure of the BV particles of AcMNPV virus via high resolution cryo-EM, and elucidated the composition and symmetrical arrangement of the apical cap of the nucleocapsid, which provided a structural basis for the DNA packaging of baculovirus. Generally, the textual description of the work is appropriate, but there are some problems with the figures. 1) There is no scale bar on any EM images or 2D averages. 2) The relative size of the panels are not well calibrated within one figure, for instance, the Extended Fig. 1C and 1D are clearly inconsistent for representing the outside and inside of the nucleocapsid. 3) The direction of rotation (clockwise? anticlockwise?) is not clear (Fig. 4C and Fig. 5C, D). To better present structural details, I suggest that the authors prepared more detailed figures with clear labels. It is advisable to have a flowchart to show the process of structural determination.

Here are some other suggestions to further improve manuscript.

Line 61: VP39 is not a nucleoprotein. It is a capsid protein, which is a component of nucleocapsid.

Lines 110-111: How do you know that GP64 is at the poles of the virion?

Lines 149-150: To clarify "the identification of 8 new proteins in the apical cap", please list all newly identified proteins here.

Line 217: Is PTP a host protein? Are there any experimental data to support its presence on virions? Can mass spectrometry detect it? What is the structural background of Ac66 interacting with PTP?

Line 341: How is the "two dsDNA moieties" related to covalently closed circular DNA (cccDNA)?

Line 378: "straight vertical 'handle' domain, connected to a 'hook' domain". Can you specify which are 'handle' domain and 'hook' domain on Fig 5?

Reviewer #3

(Remarks to the Author)

In the manuscript titled "Structure of AcMNPV nucleocapsid reveals DNA portal organization and packaging apparatus of circular dsDNA baculovirus." by Effantin et al., the authors investigate the overall structure of the nucleocapsid of *Autographa californica* multiple nucleopolyhedrovirus by Cryo-EM.

The significance and novelty of this study lies in the identification of several new proteins that have not been reported in recent structural studies. Additionally, the authors aimed to understand the DNA packaging mechanism, and the structure determination of the portal protein considered to be a contribution in this overall process. The study is comprehensive, with the authors successfully overcoming symmetry mismatches to achieve a more accurate electron density map of specific regions. The manuscript is well-written and detailed. However, a total of eight separate structures were deposited in the PDB and EMDB, a significant proportion of which have low resolution ($>4 \text{ \AA}$). This may limit confidence in model building. This issue may not be a limiting factor in this case, given the merging of proteins with different symmetries.

There are some issues that need to be addressed:

Major comments:

1. Fig. 1: Although the overall virion structure is known, it would be beneficial to include a raw micrograph of the entire virion with a scale bar. Additionally, it should be clearly indicated from which part of the virion the reconstructions in Fig. 1A, B, and C are derived.

The color-coded boxes are provided together, which makes it somewhat tedious to identify each protein in the different images. It is recommended to group the proteins with their corresponding color codes in each individual reconstruction. For example, if the basal region contains three different proteins, these should be grouped together within the basal class and assigned to the same color code.

2. The local resolution estimation maps for all reconstructions should be provided as a supplementary figure. In addition, a parallel column displaying the FSC curves for each reconstruction should be included. This is particularly important because multiple models have been constructed, and functions have been proposed based on low-resolution maps.

3. The major finding of this article is the identification of new proteins in the Cryo-EM map, which exhibit either strong or weak density. However, it can be challenging to locate these proteins within the entire virion. It is highly recommended to include a separate table listing each newly identified protein that has not been previously reported.

A schematic representation would be helpful for locating the proteins, and including a segmented map alongside the schematic would enhance clarity. This should be part of the main figure. If adding a new figure is not feasible, it could be incorporated into Figure 5.

Additionally, if model building has been performed for the newly identified proteins, it should be presented with a "fit with the map" picture with the schematic diagram.

4. The authors use the term "packaging apparatus" multiple times. In bacteriophages and other viruses, this apparatus

includes the portal protein, small and large terminases. While the manuscript provides an in-depth analysis of the portal protein, C2 plugs, Ac54 and other related structures, it is not clear which of these are collectively referred to as the "packaging apparatus."

It would be logical to reorganize the text to create a separate section dedicated to the "packaging apparatus" with subsections for its individual components. A schematic representation or proposed model would clarify the relative location of the packaging proteins in the apical region.

5. The portal protein, which exhibits C21 symmetry, is observed in two distinct classes (Extended Data Fig. 7I-K). What is the plausible explanation for this? Could the protein adopt two different conformations? Which of these classes was used for model building?

Given the high resolution of the structure, and assuming that both classes generate sufficiently good maps for model building, are there any visible differences between the models constructed from the two different classes?

Minor comments:

1. Extended Data Figure 1A: The quality of the micrograph is not very good, and there is no scale bar provided. It is recommended to replace it with a higher-quality micrograph that includes a scale bar.
2. Line 104: The DNA-filled or empty nucleocapsid in Extended Data Figure 1A should be marked with arrows for clarity.
3. Line 107: The statement "Two different...on both poles" is unclear in Extended Data Figure 1A as the separated poles are not visible. Additionally, it is difficult to understand which region is apical and which is basal in the micrograph.
4. Figure 2: In panels G, H, and I, it would be more appropriate to show the models fitted into the maps to improve clarity.
5. Extended Data Figures 7A and 7D: What is the distinction between these two figures?
6. Figure 3: In panels G, H, and I, it is necessary to show the models fitted into the maps to clearly illustrate the interactions.

Version 1:

Reviewer comments:

Reviewer #1

(Remarks to the Author)

The authors have fully addressed all of this reviewer's comments, and publication is recommended.

Reviewer #2

(Remarks to the Author)

The authors have adequately addressed the issues I raised. I support the publication of the revised paper.

Reviewer #3

(Remarks to the Author)

In the revised version of the manuscript "Structure of AcMNPV nucleocapsid reveals DNA portal organization and packaging apparatus of circular dsDNA baculovirus." by Effantin et al., the authors have addressed all major and minor comments and provided appropriate explanations. The revised article is a strong example of using the state-of-the-art Cryo-EM technique to determine the structure of previously unsolved proteins or nucleic acids within a virus particle. This article aligns well with the journal's requirements and is recommended for acceptance.

Response to Reviewer #1:

The portal structure is fascinating, and its superficial resemblance to phage portals is remarkable. It seems that it is almost certainly the portal DNA entrance and egress. However, I was not sure what the authors were suggesting regarding the mechanism of packaging. On the one hand, they speculate (probably correctly) that the virus codes for a packaging motor-type protein that only transiently docks to the portal to pump DNA into the shell, and thus is not present in the mature virus. On the other hand, they note the resemblance of A66 to the motor protein kinesin and its interaction with the phosphatase PTP, seemingly to suggest that somehow the PTP's ability to hydrolyze triphosphates provides energy to drive packaging? How do the authors envision this working, and how does it relate to the proposed transiently assembled motor? ATPase motors have distinct features easily recognized via sequence. Does the virus code for any common motor proteins (e.g. P-loop ring ATPases??)

We thank the referee for the comment. Indeed, due to their respective sequence homology with motor protein and nucleotide phosphatase we speculate that AC66 and PTP jointly participate in the motor complex responsible of pumping the genome into the shell. Because we report the structure of a mature baculovirion it is only reasonable to suggest that, similarly to other viruses, the full packaging motor is also comprising other proteins transiently assembling onto the shell and thus being unelucidated in our study. However, as suggested by the reviewer to identify other putative candidates of such transient motor complex, we blasted each 155 baculoviral proteins for ATPase motifs (P-Loop, Walker A and B...) using the conserved domain database (CDD) (DOI: 10.1093/nar/gkac1096). Out of the whole baculovirus proteome only AC66, GTA, RNL1 (RNA ligase) and Y033 showed to be part of the P-Loop containing ATPase/phosphatase superfamilies with solely GTA harboring additional Walker motif and being identified as belonging to the P-loop NTPase Additional Strand Conserved Glutamate (ASCE) superfamily. In light of the role of ASCE ATPase in DNA translocation in bacteriophages, we propose GTA as a candidate for being part of the full motor complex and modified the text accordingly line 560-567.

Minor issues:

For readers unfamiliar with baculovirus, it would help to have a schematic figure in the introduction showing the overall arrangement of the different structural components (apical, basal, etc), including which parts have known structures.

We thank the reviewer for the important comment. We have now modified Fig. 1 and introduced labels on a raw micrograph as part of the new Figure 1A panel to indicate the overall morphology of the budded virions. Additionally, the other panels have been rearranged to better depict the organisation of the AcMNPV nucleocapsid architecture and to present the molecular composition of its different subassemblies. The newly identified structures and proteins are clearly indicated with red labels and square red boxes, respectively.

Line 71 should either read "In contrast, small viruses for which virions assembly often rely on..." or, "In contrast to small viruses for which assembly relies on..."

We thank the reviewer for the remark and correction was done in the text line 71.

In the discussion, the authors might considering referencing more recent and comprehensive models of dsDNA packaging such as:

1) Pajak J, Prokhorov NS, Jardine PJ, Morais MC. The mechano-chemistry of a viral genome packaging motor. *Curr Opin Struct Biol.* 2024 Dec;89:102945. doi: 10.1016/j.sbi.2024.102945. Epub 2024 Nov 4. PMID: 39500074.

2) Woodson M, Pajak J, Mahler BP, Zhao W, Zhang W, Arya G, White MA, Jardine PJ, Morais MC. A viral genome packaging motor transitions between cyclic and helical symmetry to translocate dsDNA. *Sci Adv.* 2021 May 7;7(19):eabc1955. doi: 10.1126/sciadv.abc1955. PMID: 33962953; PMCID: PMC8104870.

3) Pajak J, Dill E, Reyes-Aldrete E, White MA, Kelch BA, Jardine PJ, Arya G, Morais MC. Atomistic basis of force generation, translocation, and coordination in a viral genome packaging motor. *Nucleic Acids Res.* 2021 Jun 21;49(11):6474-6488. doi: 10.1093/nar/gkab372. PMID: 34050764; PMCID: PMC8216284.

We thank the referee for these suggestions. The aforementioned references are now added in the discussion when proposing a packaging mechanism and mentioning that GTA was identified as a putative ASCE ATPase, most probably being part of the full packaging motor as in bacteriophages.

Response to Reviewer #2:

The authors studied the structure of the BV particles of AcMNPV virus via high resolution cryo-EM, and elucidated the composition and symmetrical arrangement of the apical cap of the nucleocapsid, which provided a structural basis for the DNA packaging of baculovirus. Generally, the textual description of the work is appropriate, but there are some problems with the figures.

1) There is no scale bar on any EM images or 2D averages.

We thank the reviewer for this comment and incorporated missing scale bars onto every micrograph, 2D class averages and underneath of figures showing large molecular assemblies forming the virion. Scale bars are thus present for all panels of Figure 1; Figure 2 A, B; Supplementary Figure 1 A-D; Supplementary Figure 4 A, C-G; Supplementary Figure 5 A, F; Supplementary Figure 6 A, C, D; Supplementary Figure 9 A, C-F; Supplementary Figure 10 A, D; Supplementary Figure 11 A-F, H, L-N and Supplementary Figure 12 A, C

2) The relative size of the panels are not well calibrated within one figure, for instance, the Supplementary Fig. 1C and 1D are clearly inconsistent for representing the outside and inside of the nucleocapsid.

We thank the referee for the remark and carefully adjusted relative size of panels within individual figure, allowing thereby to insert a single scale bar for several panel.

3) The direction of rotation (clockwise? anticlockwise?) is not clear (Fig. 4C and Fig. 5C, D). To better present structural details, I suggest that the authors prepared more detailed figures with clear labels.

We thank the reviewer for this comment and therefore included clear labels about rotation changes in Figure 4C, Figure 5C and Supplementary Figure 6D.

It is advisable to have a flowchart to show the process of structural determination.

We thank the reviewer for this comment. Flowcharts, showing Cryo-EM data processing workflows used for structure determination of the various molecular assemblies forming the BV nucleocapsid, have been elaborated and incorporated as Supplementary Figure 2, 3, 7 and 8.

Here are some other suggestions to further improve manuscript. Line 61: VP39 is not a nucleoprotein. It is a capsid protein, which is a component of nucleocapsid.

Change was made in the text accordingly line 61.

Lines 110-111: How do you know that GP64 is at the poles of the virion?

The major surface glycoprotein GP64 of AcMNPV was reported to form spike-like structures (peplomers) seen at the ends of baculovirions (doi:10.1006/viro.2001.1191). Moreover, GP64 is widely used, through fusion construct, to display heterologous epitopes at the surface of the baculovirus for altering its tropism or turning it into next generation vaccines. Validation of pseudotyped virions is commonly achieved through immunogold electron microscopy targeting the displayed epitope and showing gold particles specifically attached to the surface of the virion head (<https://doi.org/10.1371/journal.pone.0021757>, <https://doi.org/10.1007/s12250-020-00238-x>.)

Lines 149-150: To clarify “the identification of 8 new proteins in the apical cap”, please list all newly identified proteins here.

For clarity we created a new panel in Figure 1 (panel A) where the overall nucleocapsid structure organization and composition is presented next to a micrograph depicting the virion. This panel also clearly highlights the newly identified subassemblies (in red labels) and lists their protein composition (names in red square boxes).

Line 217: Is PTP a host protein? Are there any experimental data to support its presence on virions? Can mass spectrometry detect it? What is the structural background of Ac66 interacting with PTP?

PTP is encoded within the baculoviral genome (ORF1) and has been experimentally shown as being present in both BV and ODV. In the text we refer, line 548, to publication identifying it, notably by MS (DOI: 10.1128/JVI.00040-10, DOI: 10.1099/0022-1317-76-12-2941). Despite PTP crystal structure having been solved, there is no other experimental proof apart from MS analysis onto BV and ODV preps which suggest that PTP is a component of the baculovirion and could interact with Ac66. However, our observation unambiguously identified PTP as sitting on the apical anchor-1 complex and its interaction with Ac66 strongly fits AlphaFold predictions

Line 341: How is the "two dsDNA moieties" related to covalently closed circular DNA (cccDNA)?

We thank the referee for this question. Most recent understanding of baculovirus life cycle acknowledges the cccDNA genome of baculovirus replicates in host genome through rolling circles. This implies that copies of genome unit need to be circularized before being packaged in nucleocapsids (doi: 10.1007/s007050050229, doi: 10.3390/v11070595). Hence, due to the circular nature of the genome, two dsDNA strands must be introduced simultaneously into the shell as visible in our structure. Another model could involve recircularization of genome after infection and prior replication, but such is invalidated by our observation unambiguously showing two dsDNA moieties at the portal. Also measured dimensions of the pore match with the passage of two dsDNA, with p6.9 probably being involved and counteracting repulsive charges. Last but not least, when increasing the threshold in our EM map, the two dsDNA

moieties show continuous density again supporting packaging of cccDNA genome (see below, lower threshold on the right).

Line 378: “straight vertical ‘handle’ domain, connected to a ‘hook’ domain”. Can you specify which are ‘handle’ domain and ‘hook’ domain on Fig 5?

We thank the referee for the comment. Accordingly, and for better clarity, the different domains of the asymmetric unit of C21 ring are now specified on Figure 5C.

Response to Reviewer #3:

In the manuscript titled “Structure of AcMNPV nucleocapsid reveals DNA portal organization and packaging apparatus of circular dsDNA baculovirus.” by Effantin et al., the authors investigate the overall structure of the nucleocapsid of *Autographa californica* multiple nucleopolyhedrovirus by Cryo-EM. The significance and novelty of this study lies in the identification of several new proteins that have not been reported in recent structural studies. Additionally, the authors aimed to understand the DNA packaging mechanism, and the structure determination of the portal protein considered to be a contribution in this overall process. The study is comprehensive, with the authors successfully overcoming symmetry mismatches to achieve a more accurate electron density map of specific regions. The manuscript is well-written and detailed. However, a total of eight separate structures were deposited in the PDB and EMDB, a significant proportion of which have low resolution ($>4 \text{ \AA}$). This may limit confidence in model building. This issue may not be a limiting factor in this case, given the merging of proteins with different symmetries. There are some issues that need to be addressed:

Major comments:

1. Fig. 1: Although the overall virion structure is known, it would be beneficial to include a raw micrograph of the entire virion with a scale bar. Additionally, it should be clearly indicated from which part of the virion the reconstructions in Fig. 1A, B, and C are derived. The color-coded boxes are provided together, which makes it somewhat tedious to identify each protein in the different images. It is recommended to group the proteins with their corresponding color codes in each individual reconstruction. For example, if the basal region contains three different proteins, these should be grouped together within the basal class and assigned to the same color code.

We thank the referee for the comment. Figure 1 was rearranged and is now depicting more clearly the overall morphology of the virion, showing the different basal and apical complexes and their protein

content and thus helping to understand from which part the reconstructions are coming from. A raw micrograph of entire virion was included as well as scale bars. The color code remains the same and the legends are now grouped. We hope the identification of the different proteins and their localization is now clearer thanks to the new panel Figure 1A. The purpose of whole Figure 1 is to present the overall organization of the virion and the various symmetric arrangement composing it, with, as suggested, group of proteins forming such complex being labelled with the same color. This makes the figure clearer as proteins, such as Ac101, belong to various assemblies.

2. The local resolution estimation maps for all reconstructions should be provided as a supplementary figure. In addition, a parallel column displaying the FSC curves for each reconstruction should be included. This is particularly important because multiple models have been constructed, and functions have been proposed based on low-resolution maps.

In agreement with the reviewer's comment, maps for all reconstructions showing local resolution estimation are now included as Supplementary Figures 1D; 4C; 5F; 6C; 9C, D; 10D; 11D-F and 12C. In addition, FCS curves for all models vs their corresponding maps are now displayed as Supplementary figures 1E; 4B; 5B; 6B; 10B; 11G and 12B.

3. The major finding of this article is the identification of new proteins in the Cryo-EM map, which exhibit either strong or weak density. However, it can be challenging to locate these proteins within the entire virion. It is highly recommended to include a separate table listing each newly identified protein that has not been previously reported. A schematic representation would be helpful for locating the proteins, and including a segmented map alongside the schematic would enhance clarity. This should be part of the main figure. If adding a new figure is not feasible, it could be incorporated into Figure 5. Additionally, if model building has been performed for the newly identified proteins, it should be presented with a "fit with the map" picture with the schematic diagram.

We thank the reviewer and invite them to refer to our answer to their Question 1. Indeed, we introduced for clarity a new panel as Figure 1A which indicates where each individual proteins are located within the whole baculovirus architecture and underlines, with red labels and squares, the newly identified parts and their contents. To highlight the confidence of our model building "Fit with the map" pictures, are included for the newly identified proteins, amongst others, as Figures 2 G-I and Supplementary Figures 10F; 11I-K; 12D-H. In addition of Figure 1A presenting overall architecture of the virion, a schematic representation of the nucleocapsid is now introduced in Figure 6A and B for clarity.

4. The authors use the term "packaging apparatus" multiple times. In bacteriophages and other viruses, this apparatus includes the portal protein, small and large terminases. While the manuscript provides an in-depth analysis of the portal protein, C2 plugs, Ac54 and other related structures, it is not clear which of these are collectively referred to as the "packaging apparatus." It would be logical to reorganize the text to create a separate section dedicated to the "packaging apparatus" with subsections for its individual components. A schematic representation or proposed model would clarify the relative location of the packaging proteins in the apical region.

We thank the referee for the comment. As clearly stated in the text, our work presents the architecture of mature and circulating baculovirions. This is why we decided to describe the baculovirus morphology

by going through individual complex assembly following same symmetry, and not through group of proteins harboring same putative role. We think it would be too speculative and are hence cautious when postulating some proteins that could be part of the packaging apparatus. Indeed, proteins roles are only proposed within the discussion, as only being based on bioinformatical analysis and observation of our structures. Therefore, introducing a dedicated section in the main text titled “packaging apparatus” could appear as misleading to the reader and confusing within the frame of the whole nucleocapsid organization as some domains of the proteins, present in dozens of copies, are seen within different subassemblies. As suggested to clarify the location of the various proteins, we introduced in Figure 1A an overview of the virion structural organization and in Figure 6A, B a schematic representation of the baculovirus apical region used for illustrating a proposed model for DNA packaging into which symmetry mismatches between the different complexes plays a key role.

5. The portal protein, which exhibits C21 symmetry, is observed in two distinct classes (Supplementary Fig. 7I-K). What is the plausible explanation for this? Could the protein adopt two different conformations? Which of these classes was used for model building? Given the high resolution of the structure, and assuming that both classes generate sufficiently good maps for model building, are there any visible differences between the models constructed from the two different classes?

Supplementary Fig. 7I-K (now Supplementary Fig. 11 L-N in the revised manuscript) depict two slightly different positions of the C21 ring relative to the C2 plug, an outcome from 2D classification of the complete C21/C2 portal assembly (excluding the C14 assembly). Due to the limited number of particles and the absence of a common symmetry for the C21 ring/C2 plug assembly, the 2 classes couldn't be solved to high resolution. We propose the C21 ring is attached to the rest of the apical cap through linkers contained in Ac101 and Ac66. Both proteins are found in the C21 ring and in the C2 apical core. This arrangement could lead to different positions for the C21 ring relative to the C2 assembly. We hypothesize that this deliberate “loose” attachment of the C21 ring to the apical core is necessary for the role of the C21 ring in the packaging/ejection of the viral genome. The high-resolution structure of the C21 ring was obtained from multi-body refinement of the entire apical core, followed by particle subtraction to isolate C21 particles, which were then subjected to 3D classification and refinement (Supplementary Fig. 7).

Minor comments:

1. Supplementary Figure 1A: The quality of the micrograph is not very good, and there is no scale bar provided. It is recommended to replace it with a higher-quality micrograph that includes a scale bar.

The micrograph was replaced and now includes a scale bar.

2. Line 104: The DNA-filled or empty nucleocapsid in Supplementary Figure 1A should be marked with arrows for clarity.

Empty nucleocapsid in Supplementary Figure 1A is now marked with an arrow as suggested.

3. Line 107: The statement "Two different...on both poles" is unclear in Supplementary Figure 1A as the separated poles are not visible. Additionally, it is difficult to understand which region is apical and which is basal in the micrograph.

Apical and basal poles are now labelled in main Figure 1A and in Supplementary Figure 1A. It has to be mentioned that apical and basal poles were discriminated with accuracy through systematic 2D classification to increase the signal to noise ratio. In raw micrographs of complete enveloped virions, the apical/basal poles are difficult to discriminate visually because of the co-localization of other viral components at the poles (viral membrane, glycoproteins, proteins of the tegument)

4. Figure 2: In panels G, H, and I, it would be more appropriate to show the models fitted into the maps to improve clarity.

Models fitted into maps were included in panels G, H and I of Figure 2 as suggested.

5. Supplementary Figures 7A and 7D: What is the distinction between these two figures?

Supplementary Figure 7 (now Supplementary Figure 11 in the revised manuscript) was modified for adding local resolution estimation maps and the names of each map was added above for clarity. The C2 plug was best solved as a composite 3D map obtained from a multibody refinement in RELION with two different bodies. For completeness, the consensus 3D map (before multibody refinement), the two different bodies as well as the composite 3D map combining the two bodies are now shown in Supplementary Figure 11.

6. Figure 3: In panels G, H, and I, it is necessary to show the models fitted into the maps to clearly illustrate the interactions.

Models fitted into maps were included in panels G, H and I of Figure 3 as suggested.